# A moonlighting role for LysM peptidoglycan binding domains underpins *Enterococcus faecalis* daughter cell separation

Bartłomiej Salamaga[1], Robert D. Turner[1], Fathe Elsarmane[1], Nicola F. Galley [1], Saulius Kulakauskas[2] & Stéphane Mesnage [1✉]

Control of cell size and morphology is of paramount importance for bacterial fitness. In the opportunistic pathogen *Enterococcus faecalis*, the formation of diplococci and short cell chains facilitates innate immune evasion and dissemination in the host. Minimisation of cell chain size relies on the activity of a peptidoglycan hydrolase called AtlA, dedicated to septum cleavage. To prevent autolysis, AtlA activity is tightly controlled, both temporally and spatially. Here, we show that the restricted localization of AtlA at the septum occurs via an unexpected mechanism. We demonstrate that the C-terminal LysM domain that allows the enzyme to bind peptidoglycan is essential to target this enzyme to the septum inside the cell before its translocation across the membrane. We identify a membrane-bound cytoplasmic protein partner (called AdmA) involved in the recruitment of AtlA via its LysM domains. This work reveals a moonlighting role for LysM domains, and a mechanism evolved to restrict the subcellular localization of a potentially lethal autolysin to its site of action.

---

[1] School of Biosciences, University of Sheffield, Sheffield, UK. [2] Université Paris-Saclay, INRAE, AgroParisTech, Micalis Institute, 78350 Jouy-en-Josas, France.
✉email: s.mesnage@sheffield.ac.uk

*E*nterococcus faecalis form short cell chains or pairs of cells called diplococci. The distinctive morphology of this organism was reported over a century ago[1] and is one of the criteria used for the identification of this pathogen. Recent work revealed that the minimization of cell chain size is critical for pathogenesis as it negatively impacts uptake by phagocytes[2]. The formation of short cell chains requires the activity of potentially lethal enzymes called autolysins that cleave the major component of the bacterial cell wall (peptidoglycan) at the end of the cell division[3]. *E. faecalis* genomes encode many potential peptidoglycan hydrolases belonging to multigene families (e.g., 21 genes in strain V583)[4]. Three peptidoglycan hydrolases named AtlA, AtlB and AtlC have been characterized and their contribution to daughter cell separation was investigated[5]. AtlA, was shown to display *N*-acetylglucosaminidase activity whilst AtlB and AtlC are *N*-acetylmuramidases. Despite a similar domain organisation (all three enzymes have a catalytic domain fused to a C-terminal LysM domain), AtlA was shown to play a predominant role in septum cleavage[5,6]. Inactivation of *atlA* leads to the formation of long chains[5], but how AtlA is dedicated to septum cleavage remains unknown. One hypothesis that has been proposed is that the peptidoglycan structure at the septum displays a lower crosslinking index, allowing the cleavage of glycan chains by a glucosaminidase activity to cause cell separation[2].

AtlA is a multimodular enzyme. It contains an unusually long signal peptide (53 residues, hereafter referred to as an extended signal peptide region or ESPR), an N-terminal domain predicted to be intrinsically disordered, a GH73 *N*-acetylglucosaminidase domain[7] and six imperfect peptidoglycan-binding LysM repeats separated by low complexity linkers[8]. Our previous work shed light on the control of AtlA autolytic activity which relies on several independent mechanisms[2]. Glycosylation of AtlA N-terminal inhibits activity and the cleavage of this domain by the extracellular protease GelE is required for full activity[2,9]. Proteolytic cleavage of the N-terminal domain was also proposed to have an impact on the subcellular localization of the enzyme[10]. As expected, the truncation of the LysM domain had a dramatic impact on the activity of AtlA[11] and septum cleavage[2]. Both in vitro experiment with recombinant LysM domains and the analysis of strains producing AtlA with various LysM truncations indicated that LysM repeats contribute to binding in a cooperative manner[8].

Many factors modulate the activity of AtlA but little is known about the spatial control of this autolysin. Here, we sought to investigate the mechanism that underpins the preferential localization of this protein at the septum detected by immunogold staining. We report an essential role of the C-terminal LysM domain in the recruitment of AtlA at the septum before secretion and identify a partner protein involved in this process. This work therefore reveals a moonlighting role for a ubiquitous peptidoglycan binding domain produced by bacteria.

## Results

**AtlA is localized at the division septum**. We hypothesized that the prominent role of AtlA in cell separation is underpinned by the specific targeting of this enzyme to the septum. To investigate AtlA localization, this autolysin was expressed as a chromosomal *atlA-gfp* fusion under the control of its native promoter. The GFP fluorescence was mostly associated with the septum and detected at the poles (Fig. 1a), matching the AtlA distribution detected by immunofluorescence (Supplementary Fig. 1). Flow cytometry and microscopy analyses revealed a significant but moderate increase of the bacterial cell chain in the *atlA-gfp* fusion (Fig. 1b and Supplementary Fig. 2). The *atlA-gfp* strain contained an average of 3.8 cells per chain as compared to 2.3 in the parental strain

(Supplementary Fig. 2a and 2b), and the AtlA-GFP fusion was mostly detected as a full-length protein (Supplementary Fig. 2c), indicating that the fusion had a limited impact on AtlA stability and enzymatic activity.

**The signal peptide and LysM domain of AtlA are both required for septal localization of GFP fluorescence**. Previous studies revealed that both signal peptides and cell wall binding domains can play a role in the subcellular targeting of surface proteins[12–14]. Six distinct strains expressing *gfp* fusions on the chromosome, at the *atlA* locus (Fig. 1c) were constructed. They all secreted the GFP (G) via the signal peptide of AtlA (SA) or the signal peptide of AtlB (SB), eventually fused to the LysM domain of AtlA (MA) or the lysM domain of AtlB (MB). The design of the fusions containing the LysM domain of AtlB was similar to that described previously[2]. Each strain described in Fig. 1c was analysed to explore the properties of AtlA required for septal localization (Fig. 1d). Western blot analyses using anti-GFP antibodies confirmed the production of all fusions. Comparing signals contained in cell lysates (Fig. 1e) or supernatants (Fig. 1f) also provided insight into the localization of these proteins.

Fusing the GFP to the signal peptide and LysM domain of AtlA ($s_Agm_A$ strain) indicated that these two targeting signals were sufficient for its septal targeting. The lack of a LysM domain ($s_Ag$ strain) led to the accumulation of the GFP inside the cytoplasm. By contrast, the GFP fused to the canonical signal peptide of AtlB (22 residues) allowed the efficient secretion of the GFP in supernatants, with very little fluorescence associated with the cells (Fig. 1d). In both strains, the GFP was detected by immunoblotting. As expected from the microscopy analysis, GFP was exclusively associated with $s_Ag$ cells whilst this protein was entirely secreted and only detected in $s_Bg$ supernatants (Fig. 1e, f). The band corresponding to the GFP signal in the $s_Ag$ cell extracts was slightly higher than the GFP, suggesting that the signal peptide was not processed, leading to the accumulation of the GFP in the cytoplasm. Collectively, these results indicate that (i) the unusually long signal peptide of AtlA (53 residues) is associated with an inefficient secretion of the GFP across the membrane and (ii) the LysM domain is required for septal localization. Unlike what we observed in the $s_Agm_A$ strain, where the fluorescence is restricted to the septum, the GFP signal accumulated in the cytoplasm in the $s_Agm_B$ strain, indicating that AtlB LysM domain was not able to target the GFP at the septum. The same conclusion was drawn when we compared the localization of the GFP in the $s_Bgm_A$ and $s_Bgm_B$ strains; The LysM domain of AtlA was able to target the GFP at the septum whilst the LysM domain of AtlB was not. In the $s_Bgm_B$ strain, fluorescence was mostly associated with poles, not the septum.

**Truncation of the LysM domain impairs translocation across the membrane and septal localization**. Truncation of the LysM domain impairs enzymatic activity[11] and results in the increase of cell chain length[2]. We therefore hypothesized that a defect in AtlA subcellular localization could be responsible for the impaired septum cleavage and analysed the localization of AtlA derivatives with truncated LysM domains fused to the GFP. Removing LysM binding modules progressively abolished the restricted localization of AtlA at the septum. and resulted in the sequestration of fluorescent signal inside the cells (Fig. 2a). Immunoblot analyses revealed that all fusion proteins were stable and detected as full-length fusions, indicating that the accumulation of the GFP inside the cells was not due to the accumulation of the GFP resulting from proteolytic cleavage (Fig. 2b, c).

Immunofluorescence experiments indicated that reducing the number of LysM repeats led to a decrease in the amount of cell

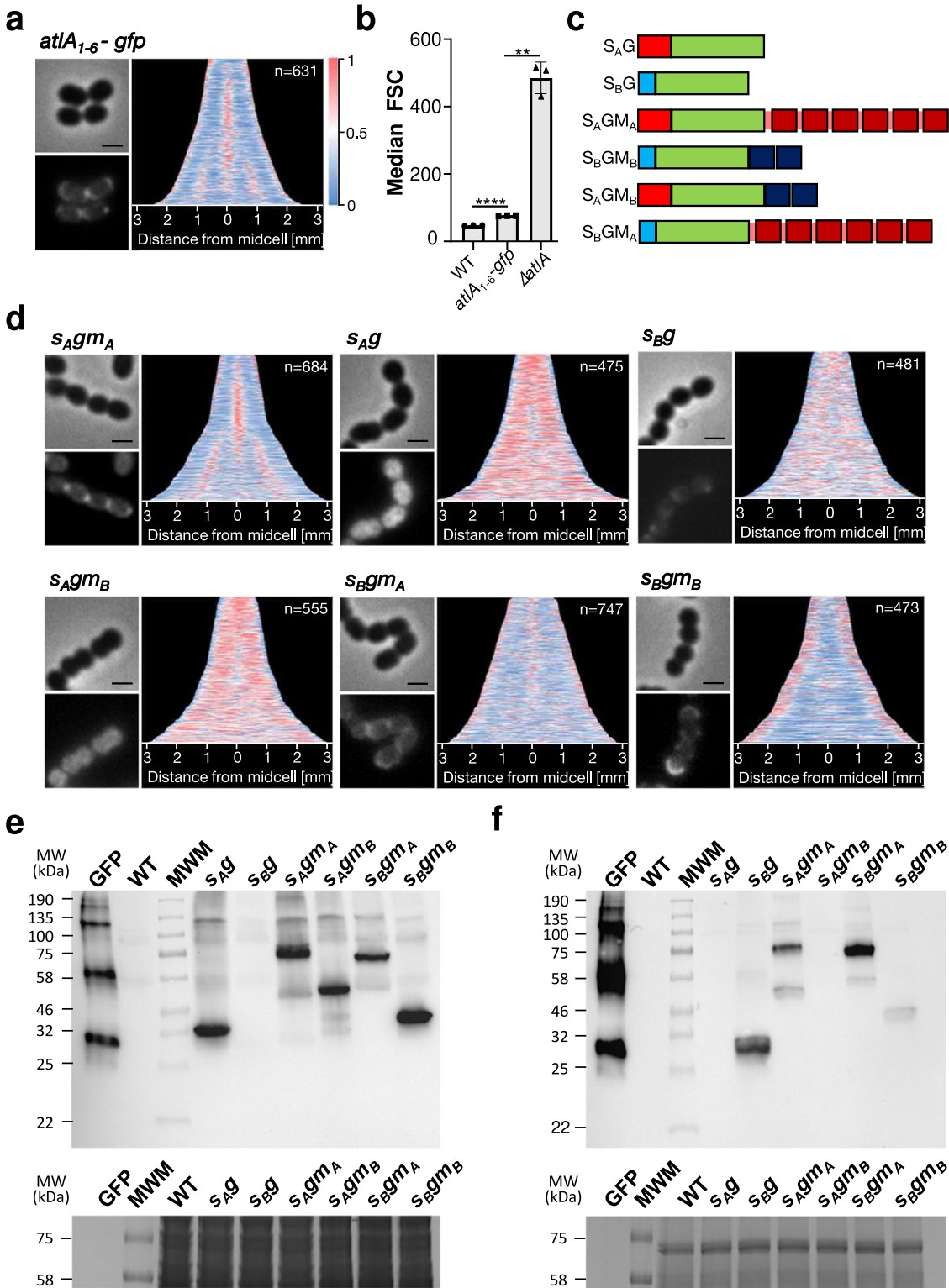

surface-displayed AtlA (Fig. 3a). Western blot analyses of cell lysates showed that AtlA derivatives with a LysM domain made of 4 binding modules or less, crude extracts contained an extra band with a molecular weight for the secreted protein (Fig. 3b, see asterisks). In agreement with this hypothesis, this higher molecular weight signal was found in cell lysates (Fig. 3c) but not in culture supernatants (Fig. 3c). This band has the expected molecular weight for the pre-protein. It is only found in the strains producing AtlA variants sequestered in the cytoplasm,

recognised by anti-AtlA antibodies. We therefore hypothesize that these bands corresponded to an accumulation of the pre-protein inside the cell before cleavage of the signal peptide. Zymogram analysis using *M. luteus* as a substrate confirmed that this band displayed peptidoglycan hydrolase activity (Supplementary Fig. 3).

**The composition of AtlA LysM domain is critical for the recruitment at the septum before translocation.** To investigate if

**Fig. 1 LysM domain from AtlA is essential for septal recruitment. a** Representative phase contrast and fluorescent images of *atlA-gfp* cells next to demographs showing AtlA-GFP localization in a population (*n* = 631). Normalized fluorescent intensity of the GFP fusion was quantified for each cell and the resulting heat maps of fluorescence were arranged according to cell length and stacked to generate the demographs using MicrobeJ. The scale bars are 1 μm. **b** Comparison of median forward scattered (FSC) light values corresponding to the cell chain lengths of WT, *atlA-gfp* and Δ*atlA* strains. Statistical analyses were carried out on results from biological triplicates using *t*-tests with Welch's correction. For JH2-2 versus *atlA-gfp*, ****$P$ = 1.42 × 10$^{-6}$, *atlA-gfp* versus Δ*atlA*, **$P$ = 0.0044. Error bars defined as standard error of mean. **c** Schematic representation of GFP fusions produced by recombinant *E. faecalis* JH2-2 derivatives. GFP is represented in green; AtlA and AtlB signal peptides are represented in red and blue, respectively; AtlA and AtlB LysM domains are represented in dark red and dark blue, respectively. **d** Representative phase contrast and fluorescent images of $s_Agm_A$, $s_Ag$, $s_Bg$, $s_Agm_B$, $s_Bgm_A$ and $s_Bgm_B$ cells next to demographs showing fluorescence distribution in a population (*n* > 450). Normalized fluorescent intensity of the GFP fusion was quantified for each cell and the resulting heat maps of fluorescence were arranged according to cell length and stacked to generate the demographs using MicrobeJ. The scale bars are 1 μm. **e, f** Immunoblotting analysis using anti-GFP antibodies probed with whole cell lysates (**e**) and culture supernatants (**f**) of $s_Ag$, $s_Bg$, $s_Agm_A$, $s_Agm_B$, $s_Bgm_A$ and $s_Bgm_B$ strains. GFP, Recombinant protein; MWM molecular weight marker. Due to the amount of GFP used as a control, several bands corresponding to multimeric forms are detected. The monomeric form of GFP is 27.8 kDa. Note that the fluorescence intensity has been normalised using the most intense signal associated with of each strain so the comparison of demographs provides a qualitative comparison.

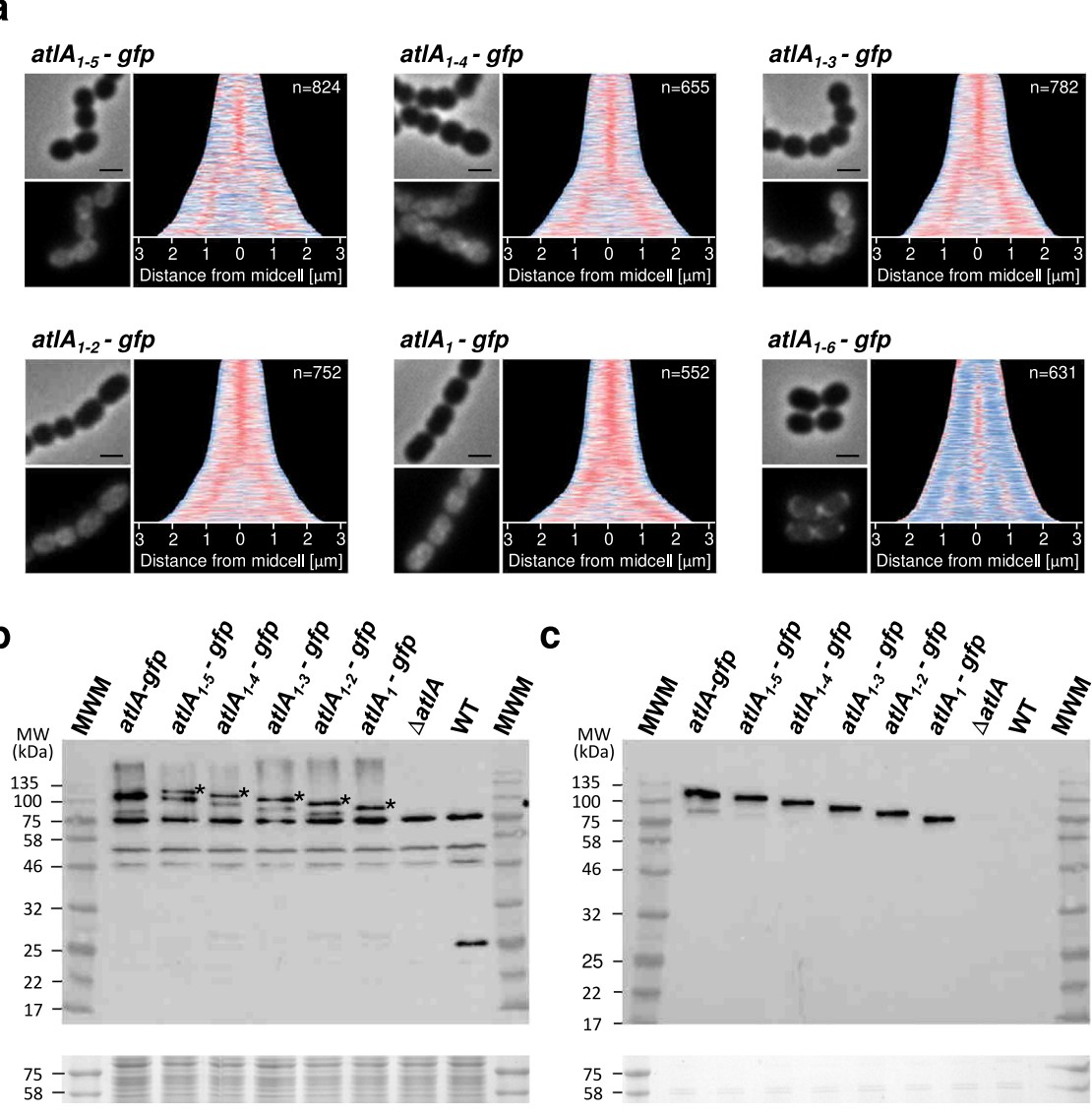

**Fig. 2 Truncation of LysM domain alters septum recruitment of AtlA-GFP fusions. a** Representative phase contrast and fluorescent images of *atlA$_{1-5}$-gfp*, *atlA$_{1-4}$-gfp*, *atlA$_{1-3}$-gfp*, *atlA$_{1-2}$-gfp* and *atlA$_1$-gfp* cells next to demographs showing AtlA-GFP derivatives localization in a population (*n* > 550). Normalized fluorescent intensity of the GFP fusions were quantified for each cell and the resulting heat maps of fluorescence were arranged according to cell length and stacked to generate the demographs using MicrobeJ. The scale bars are 1 μm. The panel with images and the demograph for *atlA-gfp* strain is copied from Fig. 1 and is used as a control. **b, c** Immunoblotting analysis using anti-AtlA antibodies probed with cell lysates (**b**) and culture supernatants (**c**). Bands corresponding to unprocessed AtlA with truncated LysM domains are indicated with asterisk.

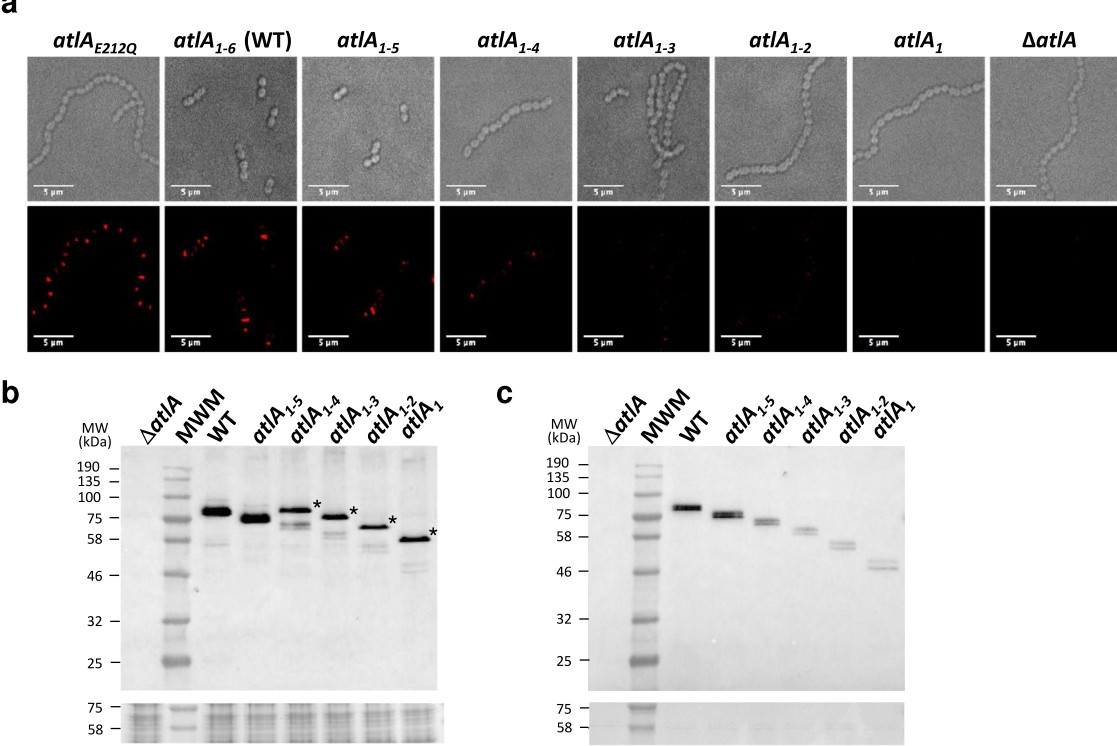

**Fig. 3 AtlA LysM truncations lead to the sequestration of AtlA inside the cell and impairs cell surface translocation. a** Bright-field and fluorescent images of *E. faecalis atlA_{E212Q}*, (a catalytic mutant with a single amino acid substitution of the catalytic Glu212 residue), JH2-2 (WT), *atlA_{1-5}, atlA_{1-4}, atlA_{1-3}, atlA_{1-2}, atlA_1* and Δ*atlA* probed with anti-AtlA serum and detected with anti-rabbit IgG antibodies conjugated with AlexaFluor647. Scale bars are 5 µm. **b**, **c** Immunoblotting analysis using anti-AtlA antibodies probed with cell lysates (**b**) and culture supernatants (**c**). Bands corresponding to unprocessed AtlA with truncated LysM domains are indicated with asterisk.

the AtlA LysM repeats were specifically required for subcellular targeting, we built strain *atlA_{1-6HB}*, producing an AtlA variant with 6 LysM repeats from the *E. faecalis* peptidoglycan hydrolase AtlB[5]. Individual LysM repeats from AtlA (LysMA1 to LysMA6) were alternatively substituted by one of the two AtlB LysM repeats (LysMB1 and LysMB2), keeping the linkers between repeats identical to those found in AtlA (Fig. 4a). Replacing the AtlA LysM binding modules by those from AtlB resulted in a septum cleavage defect; strain *atlA_{1-6HB}* formed longer cell chains, indicating that 6 LysM repeats are not sufficient for optimal activity of the enzyme (Fig. 4b). Despite the cell chain formation, the chimeric AtlA-6HB protein produced by the *atlA_{1-6HB}* strain was detected as an active, full-length protein, produced at similar levels as the wild-type AtlA protein (Fig. 4c). The modification of the LysM domain did not alter the amount of AtlA produced, its stability or the enzymatic activity detected by zymogram (Fig. 4c). Unlike the AtlA-GFP fusion, the fluorescence associated with the AtlA_{1-6HB}−GFP was less restricted to septum but also present in the cytoplasm (Fig. 4d). Similarly to the AtlA variants with a truncated LysM domain, the AtlA_{1-6HB} pre-protein was found in crude extracts. Collectively, the data presented indicated that the number of LysM repeats is not sufficient for the recruitment of this autolysin to the septum before it is translocated to the cell surface. The composition of LysM reapeats also play an important role in this process.

**Transposon mutagenesis identifies AdmA, a membrane protein contributing to the septal targeting of AtlA.** AtlA activity can be detected on agar plates containing *Micrococcus luteus* autoclaved cells as a substrate. Extracellular AtlA is able to diffuse in the agar and the hydrolysis of *M. luteus* cells forms a clear halo around colonies (Supplementary Fig. 4). We built a *Mariner*-

based transposon mutagenesis and screened ~20,000 individual colonies using this plate assay to identify mutants with altered AtlA activity. Insertions associated with an altered autolytic activity were mapped in seven independent loci (Supplementary Table 1). As expected, a large number of insertions were found in *atlA*. The other insertions were mapped in loci encoding 2 transcriptional regulators, a PTS transporter, an oligoendopeptidase and a hypothetical protein that we named AdmA for **A**tlA **d**isplay **m**utant **A**. This candidate was chosen for further characterization due to its predicted topology, which makes it a good candidate to recruit AtlA at the septum. AdmA is a predicted cytoplasmic protein of 176 amino acid protein with an N-terminal membrane anchor. We confirmed that the *admA* in-frame deletion mutant presents a long cell chain phenotype that can be partially restored upon complementation (Fig. 5a). Western blots experiments indicated that *admA* inactivation leads to the accumulation of AtlA pre-protein inside the cells (Fig. 5b). When introduced in the *atlA-gfp* genetic background, the *admA* deletion abolished the septal localization of AtlA and led to the sequestration of the GFP fusion in the cytoplasm (Fig. 5c), phenocopying truncations of the LysM domain. Complementation of the *admA* mutation with an inducible copy of the gene under the control of an inducible promoter led to the formation of shorter cell chains and the presence of AtlA-GFP foci at the septum and poles. When fused with the fluorescent protein mScarlet-I, AdmA was localized at the septum and equatorial rings (Fig. 5d).

## Discussion
Separation of daughter cells at the end of cell division has implications for host-pathogen interactions. In major pathogens such as *Staphylococcus aureus*, *Streptococcus pneumoniae* or *E. faecalis*, impaired septum cleavage leads to a decreased capacity

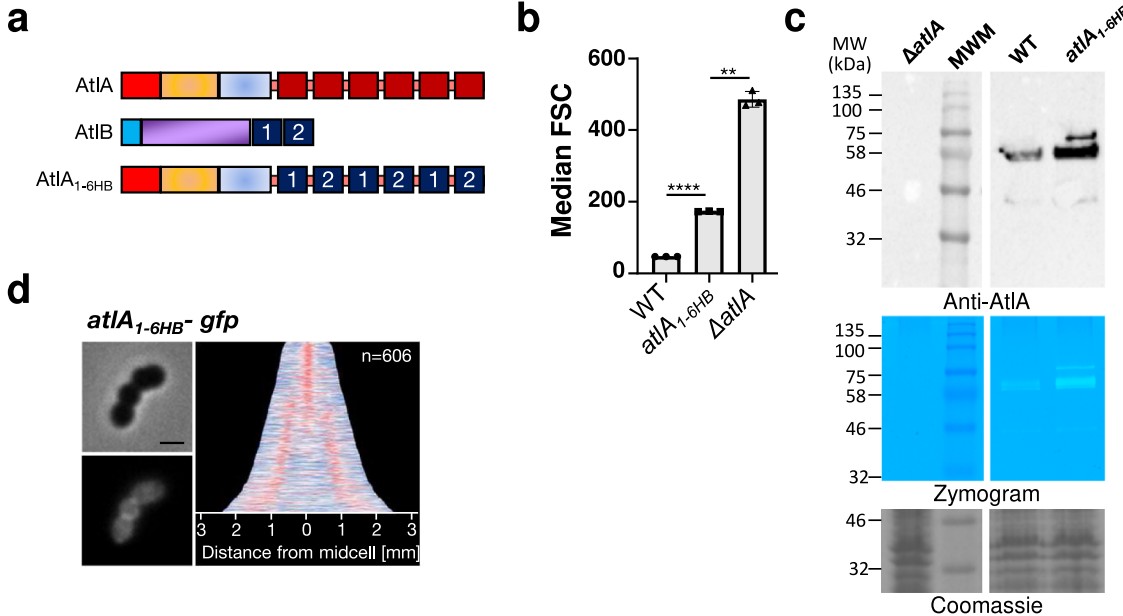

**Fig. 4 Replacing AtlA LysM repeats by AtlB LysM repeats impairs restricted localisation at the septum. a** Schematic representation of AtlA, AtlB and AtlA$_{1-6HB}$ domains organization. AtlA consist of a signal peptide (red), an N-terminal domain (orange), a catalytic domains (blue) and a LysM domain (dark red). AtlB consists of signal peptide (blue), a catalytic domain (purple) and LysM domain (dark blue). AtlA$_{1-6HB}$ consists of AtlA signal peptide (red), AtlA N-terminal and catalytic domains (orange and blue) and LysM domain made of AtlB LysM repeats (dark blue). **b** Comparison of median forward scattered (FSC) light values corresponding to the cell chain lengths of WT, atlA$_{1-6HB}$ and ΔatlA strains. Statistical analyses were carried out on results from biological triplicates using t-tests with Welch's correction. JH2-2 versus atlA$_{1-6HB}$ ****$P = 2.3 \times 10^{-8}$; atlA$_{1-6HB}$ versus ΔatlA, **$P = 0.0016$. Error bars defined as standard error of mean. **c** Immunoblotting analysis of cell lysates of ΔatlA, JH2-2 (WT) and atlA$_{1-6HB}$ strains probed with anti-AtlA antibodies. Bands corresponding to expected size of AtlA (76 kDa) were detected. The extra band corresponding to the unprocessed AtlA was detected in the atlA$_{1-6HB}$ cell lysate, indicating an increased amount of the pre-protein. Bottom panel shows AtlA activity on a zymogram containing M. luteus autoclaved cells as a substrate. **d** Representative phase contrast and fluorescent images of atlA$_{1-6HB}$-gfp cells next to demographs showing AtlA$_{1-6HB}$-GFP localization in a population (n = 606). Normalized fluorescent intensity of the GFP fusion was quantified for each cell and the resulting heat map of fluorescence was arranged according to cell length and stacked to generate the demographs using MicrobeJ. Scale bar is 1 μm.

to evade the innate immune response and an attenuated virulence[2,15–17]. In *E. faecalis*, the *N*-acetylglucosaminidase AtlA is dedicated to septum cleavage[5]. Here, we reveal the mechanism that controls the spatial distribution of this enzyme at the division site. We demonstrate an unexpected dual role for the cell wall binding domain of AtlA: it promotes the recruitment of AtlA at the septum and binds to the peptidoglycan outside of the cell after secretion, allowing cell wall hydrolysis and daughter cell separation.

AtlA sequence and domain architecture are unusual. It contains a long signal peptide (53 residues) and a multimodular LysM domain, both playing a critical role for septal localization. Elongated signal peptides have only been studied in proteins from Gram-negative bacteria[18–21]. The two model systems described in the literature include the signal peptide of the serine protease autotransporter (Pet) of the Enterobacteriaceae and the filamentous hemagglutinin (FHA) produced by *Bordetella pertussis* (52 and 71 residues, respectively). Both proteins contain an N-terminal extended signal peptide region (ESPR) (23 amino acids in Pet, 25 in FHA) non-essential for secretion that delays the translocation across the cytoplasmic membrane by an unknown mechanism. Our Western blot analyses and fluorescence microscopy experiments support the idea that AtlA ESPR plays a similar role in *E. faecalis*. As opposed to the canonical signal peptide present in the peptidoglycan hydrolase AtlB, AtlA signal peptide containing an ESPR is unable to drive efficient GFP secretion and the preprotein accumulates inside the cells (Fig. 1e, f). A similar result is observed when the signal peptide of AtlA is secreting the GFP fused to the LysM domain of AtlB. Although we cannot formally exclude that fusing the GFP to the signal

peptide of AtlA is causing protein aggregation in the absence of the LysM domain of AtlA, we propose that the cytoplasmic distribution of the SA-GFP and SAGMB fusions is due to the fact that these are not recruited at the septum. Restricted septal localization of GFP fusions was only observed when both the signal peptide and LysM domain of AtlA were combined, prompting us to investigate the role of the LysM domain in subcellular targeting. Truncation of two LysM repeats, or more was associated with the progressive loss of surface localization and the accumulation of a high molecular band corresponding to the pre-protein inside the cells. Collectively, these results suggest that optimal secretion requires both AtA signal peptide and the full-length LysM domain. *E. faecalis* genome encodes 11 putative proteins containing a LysM domain made of either a single or tandem LysM repeats. AtlA is the only protein that contains six LysM repeats. This unique LysM domain architecture is critical for the enzymatic activity of this enzyme[2,11]. We showed that the presence of six LysM repeats in AtlA is not sufficient to underpin septum cleavage and promote the restricted localization of the GFP to the septum. Swapping all AtlA LysM repeats with those from another protein impaired the restricted septal localization of AtlA (Fig. 4), indicating an unforeseen contribution of this domain during the intracellular trafficking of the protein to the septum.

Limited information is available on the localization of *E. faecalis* secretion apparatus but immunogold labelling experiment indicated that SecA is located at the septum[22]. Translocation across the membrane therefore requires a recruitment of proteins to this site. Using a functional screen, we identified a predicted bitopic protein, AdmA with a cytoplasmic C-terminal extension required for the

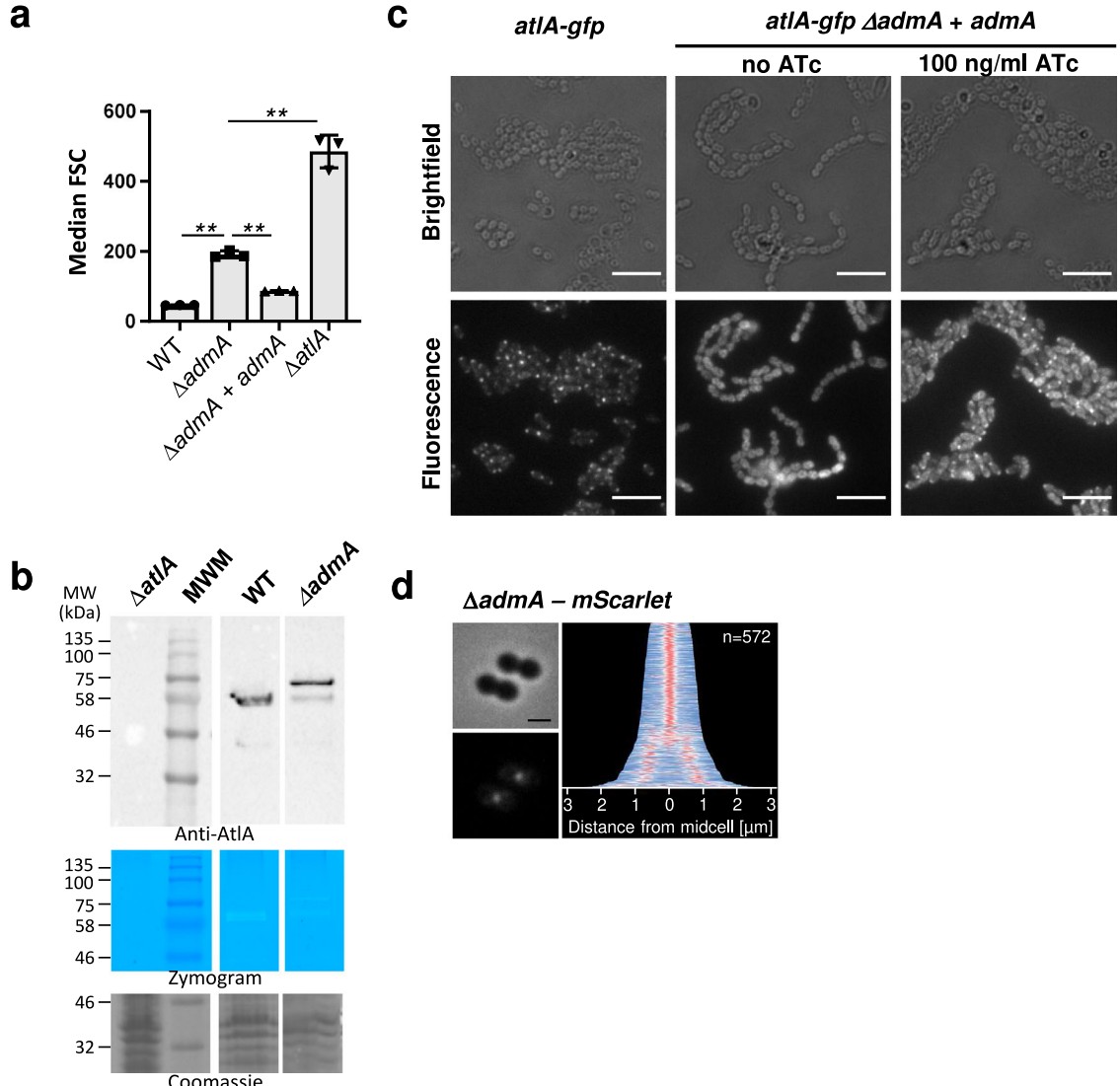

**Fig. 5 AdmA is essential for AtlA recruitment at the septum and is preferentially localised at the septum. a** Comparison of median forward scattered (FSC) light values corresponding to the cell chain lengths of WT, ΔadmA, ΔadmA + admA (complemented deletion strain) and ΔatlA strains. Statistical analyses were carried out on results from biological triplicates using t-tests with Welch's correction. JH2-2 versus ΔadmA, **P = 0.0015, ΔadmA versus ΔadmA/admA+), **P = 0.0028, ΔadmA versus ΔatlA, **P = 0.0068. Error bars defined as standard error of mean. **b** Immunoblotting analysis using whole cell lysates of ΔatlA, JH2-2 (WT) and ΔadmA strains probed with anti-AtlA antibodies. AtlA activity was assayed by zymogram, and a loading control (Coomassie) is shown. **c** Bright field and fluorescent images of the atla-gfp and ΔadmA complemented derivative (atlA-gfp ΔadmA + admA). Strains were grown int the presence of 100 ng/ml of anhydrotetracycline (Atc) to induce the expression of admA.d. Representative phase contrast and fluorescence images of the adma-mScartet cells next to demographs showing AtlA-GFP in a population (n > 570). Normalized fluorescent intensity of the GFP and mScarlet fusions were quantified for each cell and the resulting heat maps of fluorescence were arranged according to cell length and stacked to generate the demographs using MicrobeJ. The scale bars are 5 μm (**c**) and 1 μm (**d**).

localization of AtlA at the septum. The restricted localization of AdmA at the septum led us to hypothesize that the AtlA LysM domain could be involved in a direct interaction with this protein. Isothermal titration Calorimetry assays using either full length recombinant AtlA or the LysM domain alone did not reveal any interaction with AdmA (residues 26-172) (Supplementary Fig. 5). Similarly, a split luciferase assay in *E. faecalis* failed to show any direct interaction between AtlA or its LysM domain and AdmA (Supplementary Fig. 6). These experiments thus suggest that the recruitment of the LysM domain by AdmA is likely to involve one (or more) co-factor(s) that remains to be identified. Since septum hydrolysis occurs at the end of cell division, it is tempting to assume that components of the late division machinery are involved, providing temporal control for daughter cell separation.

Our previous work revealed that AtlA activity in under the control of post-translational modifications including the glycosylation of its N-terminal domain (that inhibits activity) and proteolytic cleavage by extracellular proteases[2]. Collectively, our work therefore revealed that the autolytic activity of AtlA is under the control of several mechanisms that act synergistically. These mechanisms prevent cell lysis and optimize septum cleavage activity via both temporal and spatial control.

## Methods

**Bacterial strains, plasmids and growth conditions**. Bacterial strains and plasmids used in this study are described in the Supplemental Experimental Procedures. *E. coli* was grown at 37 °C in Brain Heart Infusion (BHI) broth or agar 1.5% (w/v) supplemented with 200 μg/ml erythromycin for pGhost9 derivatives. *E. faecalis* strains were grown in BHI broth or agar at 37 °C, unless otherwise stated.

When necessary, the medium was supplemented with 30 μg/ml erythromycin and anhydrotetracycline at a final concentration of 100 ng/ml.

**Construction of pGhost derivatives for allele replacement**. All plasmid sequences described in Supplementary Table 2 are available upon request. Supplementary Table 3 contains all the information concerning plasmid construction and the sequences of synthetic DNA fragments used in this work. The sequence of the chimeric proteins encoded by these plasmids is described in Supplementary Fig. 7.

*Signal peptide*-gfp fusions. Plasmid pGSAG is a derivative of plasmid pGHH0799[5] built using Gibson assembly. A PCR fragment encoding the GFP was amplified with oligonucleotides SAG_G_Fw and SAG_G_Rev and cloned into pGHH0799 digested with NcoI and BglII. pSAGMA and pSAGMB were constructed by restriction cloning. Fragments encoding LysM domains of AtlA or AtlB were amplified from *E. faecalis* JH2-2 chromosomal DNA with oligonucleotides SAG-MA_Bgl_Fw + SAGMA_Bgl_Rev and SAGMB_Bgl_Fw + SAGMB_Bgl_Rev, respectively. The PCR products were digested by BglII, purified and cloned into plasmid pSAG cut with BglII and dephosphorylated.

To build pSBG, we replaced the EcoRI-NcoI fragment from pSAG by synthetic DNA fragment corresponding to the 596 nucleotides upstream of AtlA start codon followed by a fragment encoding the signal peptide of AtlB (22 residues). The construction of pSBGMA and pSBGMB was carried out by restriction cloning of a BglII fragment encoding the LysM domains of AtlA or AtlB, respectively into pSBG cut with BglII and dephosphorylated as previously described for pSAGMA and pSAGMB.

*atlA-6HB fusion.* For plasmid pGatlA1-6HB, a synthetic DNA fragment cloned in pUC-GW-Amp and flanked by XhoI and EcoRI was inserted in pGhost9 by restriction cloning. The final insert contained the catalytic domain of AtlA fused to 6 LysM repeats alternating repeats 1 and 2 of AtlB (separated by the same linkers as those in AtlA LysM domain) and 250 bp of 3′ untranslated region downstream of AdmA stop codon.

*lysM-gfp fusions.* All pGhost derivatives encoding AtlA-GFP fusions were built using a similar strategy involving a Gibson assembly with 3 DNA fragments: (i) a PCR fragment encoding the catalytic domain followed by 6 LysM repeats, amplified with primers atlA_G_Fw and atlA1-X_G_Rev from JH2-2 genomic DNA (where X represents the number of LysM repeats in the construct), (ii) a fragment encoding a short linker and the eGFP reporter followed by 250 bp of the untranslated region downstream of AtlA stop codon, amplified from a synthetic DNA using primers XL_GFP_G_Fw and L-GFP_G_Rev (where X represents the number of repeats in the construct) and (iii) pGhost digested by XhoI and KpnI.

Plasmid pGatlA1-6HB-GFP was cloned by replacing the BglII-XhoI fragment of pGatlA1-6HB by a synthetic gene flanked by the same restriction sites and cloned in pUC-GW-Amp (Genewiz). The final insert contained the catalytic domain of AtlA fused to 6 LysM repeats alternating repeats 1 and 2 of AtlB and 420 bp of 3′ untranslated region downstream of AdmA stop codon.

*admA complementation.* The open reading frame encoding full length AdmA was amplified using oligonucleotides AdmA_G_Fw and AdmA_G_Rev and cloned by Gibson assembly in pTETH digested by NcoI and BamHI.

*admA deletion.* Two chromosomal fragments corresponding to the upstream and downstream regions (675 and 530 bp, respectively) flanking the *admA* gene were amplified by PCR with oligonucleotides AdmA_H11 + AdmA_H12 and AdmA_H21 + AdmA_H22. PCR fragments were purified and fused by overlap extension PCR[23] using oligonucleotides AdmA_H11 + AdmA_H22 containing and cloned in pGhost9 by restriction cloning using XhoI and EcoRI.

*atlA-mScarlet fusion.* A synthetic DNA fragment cloned in pUC-GW-Amp and flanked by XhoI and EcoRI was inserted in pGhost9 by restriction cloning. The final insert contained the full length admA gene followed by a linker, the mScarlet gene and 500 bp of 3′ untranslated region downstream of *admA* stop codon.

**Mariner based transposon mutagenesis**. The *Mariner*-based transposon mutagenesis was carried out in *E. faecalis* OG1RF as previously described[24,25]. OG1RF was electroporated with plasmid pZXL5. Transformants were selected on plates containing chloramphenicol and gentamicin at 28 °C and grown to mid-exponential phase at 28 °C before transposition was induced by addition of nisin (25 ng/ml). The culture was then transferred to 42 °C overnight to counter-select the replication of the plasmid. The library was amplified by growing the cells at 42 °C in the presence of gentamicin overnight.

**screening of transposon mutants with altered AtlA activity**. The transposon library was screened on BHI-agar plates containing gentamicin and *M. luteus* autoclaved cells (SIGMA) as a substrate (OD$_{600}$ = 3). AtlA activity was detected as a clear halo around the colonies. In the absence of AtlA, no activity is detected on

*M. luteus* plates, indicating that this assay is specific for this enzyme (Supplementary Fig. 4a).

**Mapping transposition sites**. Transposon insertion sites were mapped using two divergent primers (Mar_up and Mar_dn) on the transposon by reverse PCR. Genomic DNA was extracted using the Promega Wizard kit and digested at a concentration of 4 ng/μl in a volume of 30 μl by SspI. Restriction products were diluted to 1 ng/μl and self-ligated for 16 h at 16 °C after the addition of 100U of T4 DNA ligase (NEB). Three microliters of the ligation product were used as a template for PCR amplification using oligonucleotides Mar_up and Mar_dn. PCR products were purified by gel extraction and sequenced with the T7 rpimer. The insertion site was defined as the first nucleotide of the *E. faecalis* OG1RF genome downstream of the inverted repeat sequence flanking the transposon.

**Constructions of *E. faecalis* mutants**. Isogenic derivatives of *E. faecalis* JH2-2 were built by allele exchange as previously described[5]. Briefly, pGhost derivatives were introduced into JH2-2 by electroporation and transformants were selected at 28 °C on BHI plates with erythromycin. Single crossover recombination was induced by incubating transformants at a non-permissive temperature (42 °C) in the presence of erythromycin. The second recombination event leading allelic exchange was obtained following 5 serial subcultures at 28 °C without erythromycin. The last subculture was plated at 42 °C without erythromycin and clones harbouring a double crossover mutation were identified by PCR and Southern blot hybridization.

**Protein preparation from *E. faecalis* cultures**. Proteins from supernatants were precipitated from exponentially growing cultures (OD$_{600}$~0.4) by the addition of 10% (m/v) trichloroacetic acid. After 10 min on ice, proteins were spun down (15,000 × g, 10 min at room temperature), washed in 100% acetone, dried and resuspended in SDS-PAGE loading buffer (1 ml/equivalent OD$_{600}$ = 50). Volumes corresponding to 100 μl of supernatants were loaded on SDS-PAGE.

For crude extracts, cells from 40 ml of exponentially growing culture (OD$_{600}$~0.4) were spun, washed in PBS, resuspended in 750 μl of PBS and transferred to a tube containing 250 μl of glass beads (100 μm diameter, Sigma). Cells were mechanically disrupted using a FastPrep device (six cycles of 40 s at maximum speed with 5 min pauses between cycles). Loading buffer was added to the protein samples and 10 μl of the cell lysates were separated on a 11% SDS-PAGE.

**Western blot detection of AtlA**. Proteins were transferred to a nitrocellulose membrane. After a blocking step for 1 h at room temperature in Tris buffer saline (TBS, 10 mM Tris-HCl pH7.4, 150 mM NaCl) supplemented with 0.025% tween-20 v/v) and 2% milk (m/v), the membrane was incubated with rabbit polyclonal anti-AtlA antibodies raised against the catalytic domain of AtlA (1:1000 dilution) or polyclonal anti-GFP antibodies (1:2000 dilution). Proteins were detected using goat polyclonal anti-rabbit antibodies conjugated to horseradish peroxidase (Sigma) at a dilution of 1:20,000 and clarity Western ECL Blotting Substrate (BioRad).

**Detection of AtlA activity by zymogram**. Proteins from supernatant were separated on a 11% SDS-PAGE containing *M. luteus* autoclaved cells as a substrate (final OD$_{600}$ = 3). After electrophoresis the gel was rinsed in distilled water and proteins were renatured at 37 °C in a buffer containing 50 mM Tris-HCl (pH 7.5) and 0.1% (v/v) TritonX-100.

**Flow cytometry analysis of cell chains length**. Cells were grown overnight without agitation at 37 °C. Cells were diluted 1:100 into fresh broth (OD$_{600}$~0.02) and grown in standing cultures to mid-exponential phase (OD$_{600}$~0.2–0.4). Bacteria were diluted 1:100 in filtered phosphate buffer saline and analysed by flow cytometry using Millipore Guava Easy Cyte system. Light scatter data were obtained with logarithmic amplifiers for 2500 events.

**Light and fluorescent microscopy**. Cells were grown to mid-exponential phase (final OD$_{600}$~0.3) and fixed with 1.6% paraformaldehyde in PBS for 30 min at RT. After fixation, bacteria were washed twice in distilled water and mounted onto poly-L-lysine coated slides and imaged using a DeltaVision deconvolution microscope equipped with an UplanSApo 100x oil (NA 1.4) objective and a Photometrics Coolsnap HQ CCD camera. ImageJ software was used to optimize contrast and to count the numbers of cells per chain. For immunofluorescence experiments, fixed cells were blocked in phosphate buffer saline containing 0.1% (v/v) Tween-20 (PBS-T) and supplemented with 2% (w/v) bovine serum albumin for 1 h at room temperature. Cells were washed in blocking solution and incubated with primary anti-AtlA antibodies at a 1:250 ratio in blocking solution overnight at 4 °C. Cells were washed three times in 0.5 ml of PBS-T and incubated in the presence of goat anti-rabbit secondary antibodies conjugated with AlexaFluor647 resuspended in a ratio of 1:400 in blocking solution for 2 h at room temperature. After incubation, cells were washed three times in PBS-T and resuspended in 100 μl of the same buffer.

**ITC**. Both LcpI and AdmA were expressed as recombinant C-terminally His-tagged proteins using pET2818 as an expression vector. Recombinant LysM corresponded to residues 338-737 and AdmA to residues 27–173, followed by the octapeptide GSHHHHHH. Protein production was induced in 1.5 l of exponentially growing BL21(DE3) cells with 0.5 mM Isopropyl β-D-1-thiogalactopyranoside. After 4 h, cells were harvested, resuspended in 40 ml of buffer A (50 mM Tris-HCl + 500 mM NaCl) and mechanically disrupted using a French press (one passage at 1200 psi). The soluble fraction was purified by immobilized metal affinity chromatography on a 5 ml resin column using buffer A and a gradient to 100% of buffer A supplemented with 500 mM imidazole over 20 column volumes. Fractions containing the protein of interest were pooled and purified on a Superdex75 HiLoad HR 26/600 column equilibrated with the ITC buffer (in 50 mM Tris pH 8.5, 150 mM NaCl containing 0.05% (v/v) tween-20). Fractions containing pure protein were pooled and concentrated to 625 μM (LysM) or 67.5 μM (AdmA) using their extinction coefficient to determine protein concentration. All relevant information concerning ITC are provided with Supplementary Fig. 5.

Isothermal microcalorimetry was carried out using a nano ITC instrument (TA Instruments). A 625 μM LysM solution was titrated into a 67.5 μM AdmA solution. Sequential injections corresponding to 2 μl of purified LysMA (625 μM) were made into 182 μl of AdmA protein solution (67.5 μM). Control injections corresponded to 2 μl of LysMA injected into a buffer solution (50 mM Tris pH 8.5, 150 mM NaCl containing 0.05% (v/v) tween-20).

**Split luciferase assays**. The plasmids used for the split luciferase assays are described in Supplementary Fig. 6. Their sequences were determined by Plasmidsaurus and are available upon request. Briefly, overnight cultures of *E. faecalis* were diluted to an $OD_{600}$ of 0.02 and grown for 2 h at 37 °C. The expression of the protein fusions was induced by addition of anhydrotetracycline at a final concentration of 100 ng/ml and growth at 37 °C for 1 h. The cell density was normalised to an $OD_{600}$ of 0.1 and luciferase activity was measured using the Nano-Glo® luciferase assay according to manufacturer's instructions (Promega N1110). The assay was performed in three biological repeats using a white 96-well plate and a HiDex Sense plate reader (Hidex).

**Statistics and reproducibility**. For flow cytometry experiments, biological triplicates were used. All data are presented with error bars defined as standard error of mean. Statistical analyses were performed using an unpaired, two-sided Student's *t*-test to calculate *P* values with Prism v.8.4.3 (GraphPad Software).

## Data availability
All plasmid sequences are available upon request. Source can be obtained from the ORDA repository (available at https://doi.org/10.15131/shef.data.22306045). Uncropped and unedited blot images and gels are provided in Supplementary Fig. 8. Raw data corresponding to flow cytometry experiments in Figs. 1b, 4b and 5a are provided in Supplementary Data 1.

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

## Acknowledgements
This work was supported by a BBSRC grant (BB/N000951/1).

## Author contributions
B.S. and S.M. conceived and planned the study. B.S., F.M.E., N.F.G., S.K. and S.M. performed the experiments. B.S., R.D.T., S.K. and S.M. analysed the data. B.S. and S.M. prepared the figures. B.S. and S.M. wrote the manuscript and all authors edited and approved the final manuscript version.

## Competing interests
The authors declare no competing interests.
