## [Peer Review File · Communications Biology]

Reviewers' comments:

Reviewer #1 (Remarks to the Author):

The manuscript by Salamaga et al reports a moonlighting role for LysM peptidoglycan binding domains. They showed that the LysM domain of the N-acetylglucosaminidase AtIA is essential for the localization at the division site in *Enterococcus faecalis*. Using transposon mutagenesis, they also identify a membrane-bound protein AdmA that is required for the recruitment of AtIA at the septum.

Comments to the Authors:

1. Lines 70-71: The authors say that the atIA-gfp fusion had a minor impact on AtIA activity. However, according to the results shown in Fig. 1B, the atIA1-6-gfp strain showed a significant increase in cell chain length ($P = 1.42 \times 10^{-6}$). It would be a much better demonstration if the authors show the actual cell length of these strains. Does the GFP fusion change the stability of AtIA?
2. Lines 80-82: it is unclear why the authors used AtIB to construct these GFP fusion strains. What's the relationship between AtIA and AtIB? What's the function of AtIB?
3. Fig. 1E and 1F, why the GFP protein shows several bands in the first lane?
4. Line 95: it is unclear why the authors claim that the GFP fluorescence in the SBGMB strain was associated with the "old" peptidoglycan.
5. Fig. S2: Authors should provide the immunoblotting analysis with culture supernatants. What does the atIAE212Q strain mean in the Fig. S2A?
6. Line 126: Fig. S3E is missing in the supplementary material.
7. Lines 130-138: It would be nice to see the halos of these transposon mutants. Why the authors have chosen AdmA for further investigation, in particular that this is a hypothetical protein.
8. Lines 151-154: It has not been established that the LysM domain promotes the recruitment of AtIA at the septum via AdmA. Only a correlation is made. In the absence of further evidence, they should tread more lightly here.
9. Fig. 3C: the complementation of the Δ admA should be also performed.
10. Several figures in the supplementary material should be moved to the main text, such as Fig. S2, Fig. S3.
11. Lines 160-164: the authors indicate that the AtIA protein possesses an ESPR. This information should be included in the introduction section.
12. The methodology for transposon mutagenesis and zymogram analysis should be much more detailed.
13. There are also many mistakes along the manuscript. For instance:
 - 1) Line 59: the appropriate reference should be provided.
 - 2) Line 62: the reference 10 should be removed from this sentence.
 - 3) Line 68: the authors should clarify the strain atIA1-6-gfp.
 - 4) "localisation" should be "localization".
 - 5) The authors used "JH2-2" in Fig. 1B and Fig. 3, but "WT" in Fig. 1E and 1F.
 - 6) Line 111: *M. lysodeikticus* or *Micrococcus luteus*?

- 7) Line 123: the result for zymogram analysis is not shown in Fig. S3C.
- 8) Fig. S2: What do the arrows mean?
- 9) Table S2: the description for each strain or plasmid should be unique.

Reviewer #2 (Remarks to the Author):

Review: A moonlighting role for LysM peptidoglycan binding domain underpins daughter cell separation.

This work sought to understand how AtIA is localized to the septum. Salamanga et al. claimed that the LysM domain of AtIA is required for its septal localization. In addition, the authors suggested AtIA is recruited to the septum before secretion (lines 31 and 64). Also, they identified six genes (Table S1) that when inactivated by a transposon, led to a decrease in AtIA function. Among them is AdmA, a transmembrane protein that is seemingly required for AtIA secretion and its septal localization. From these results, the author concluded that the autolytic activity of AtIA is "under the control of several mechanisms that act synergistically" (line 196). This claim, and several others, are nevertheless not substantiated sufficiently by the data presented. Furthermore, several alternative explanations of the results were not fully addressed. While the preliminary findings are interesting, some experiments lack the key controls required to draw definitive conclusions.

Major concerns

-Fig. 1C to F: This set of experiments attempted to show that the LysM domain of AtIA is sufficient to localize it to the septum. The authors fused the signal peptides of AtIA or AtIB to the N-terminus of GFP. Then, the LysM domains from these two proteins were added to the C-terminus of the GFP fusion. However, the "sAg", "sAgmB", and "sBgmB" fusions are not secreted and thus they could be non-functional or misfolded. These constructs are not interpretable. If I removed them from the analysis, comparing "sAgmA" and "sBgmA" suggests the signal peptide played an important role in the septal localization of AtIA, not the LysM domains.

-It is unclear why AtIB is relevant, except they are both autolysins and contain LysM repeats. There is not sufficient background about AtIB to justify why it is included in the study.

-Fig. 2: Are these constructs secreted? Is there any evidence that shows that the constructs are not degraded and produced at similar levels?

- If AtIA is not secreted (Fig. 3B), the cells should fully phenocopy Δ AtIA (Fig. 3A), but they are not. Also, there is no complementation for Fig. 3C. And the link between AdmA and the LysM repeats, if any, is not substantiated. Is there any evidence indicating the loss of AtIA is the reason why the Δ admA mutant forms chains?

-The authors mentioned truncating the LysM repeats reduced the amount of surface displayed AtIA (Fig. S2A), but such truncation did not affect its localization (lines 102-104). This result contradicts their major conclusion that LysM is required to recruit AtIA to the septum. In Fig. S2B, it is unclear why whole cultures were loaded rather than cell-free supernatant. The speculation that the top band is the preprotein of AtIA, which is crucial to their claim, has not been fully tested.

-Fig. S3D confirmed my concern that "sAgmB" in Fig. 1D is not interpretable. Unlike "sAgmB", the full-length AtIA protein with the AtIB LysM domain is septal localized. Is AtIB targeted to the septum?

-Fig. S5C, why AtIA alone and Δ lysM +AdmA generate luminescence?

-Line 94-95: what is the evidence that suggests "sBgmB" is associated with old peptidoglycan? Protein

aggregates can often be directed to the cell pole as well.

-The Materials and Methods section is unacceptably brief. There is no description of the immunofluorescent microscopy and the preparation of the whole cell. Also, the transposon mutagenesis and insertion mapping had a single-line description. What is the strain used to construct the transposon mutants? How was the screen done? How many colonies were screened? The ITC and split luciferase assays were not described sufficiently. How were the proteins purified? ITC should be Fig. S4, not S5.

Minor comments:

-Figure 1E and 1F: Is lane 3 the protein ladder? What is GFP? Is it recombinant GFP loaded as a positive control?

-Line 101: The immunofluorescent microscopy of AtIA was used to show that it is surface-exposed. What is the basis of this? The loss of signal in the LysM mutants can be caused by the inability to bind peptidoglycan, not necessarily means that AtIA is not surface exposed.

-Line 103: The signals in Fig S2A are weak (atIA1-3, atIA1-2, and atIA1). How could the authors conclude the localization of these proteins is not altered?

-Figure S2A: Is atIAE212Q a catalytic site mutant? It is not mentioned in the text. Where is E212?

-Figure S2 legends: Typo of (E), should be (D).

-There is no evidence of AdmA and AtIA colocalizing. Thus, the title of Figure 3 should be changed.

-What is the correlation between FSC and the chain length?

-Line 273: Please specify what is TCA.

Reviewer #3 (Remarks to the Author):

The manuscript by Salamaga et al. describes a novel function for LysM domains (typically associated with peptidoglycan binding) as an intracellular septal recruitment signal, likely by the newly-identified AdmA protein. This is pretty straight-forward work, but I have substantial comments on data organization and presentation.

1. Please keep *E. faecalis* in the title. This does not seem to be a general, well-conserved function, but rather specific to *E. faecalis*.
2. The introduction could use some more information on AtIA biochemistry. How does a glucosaminidase contribute to septal cleavage?
3. Fig. 1A: Is atIA1-6 the same as wt AtIA? This is not clear. In Fig. 1C, please add labels to the color to make it more clear to the reader.
4. The entire description of the truncated constructs (line ~81 – 95) is very difficult to follow and should be streamlined. Please always mention the identity of the constructs (what is sagmb stand for? The reader has to go back and forth between figure legend and the text to make sense of it).
5. Line 95: There is no evidence of old peptidoglycan. Perhaps change to "pole"
6. Line 99 states "removing LysM...abolished restricted localization..." and in line 103 "interestingly, it did not alter the localization of this protein to the septum". This seems like a contradiction. Also, in Fig. S2A, there is no visible fluorescence in half of the images, so this statement is difficult to evaluate.
7. Zymogram analysis: line 111 states that "*M. lysodeikticus*" was used. This is unclear. Was PG from that organism isolated? Did they use a whole cell in the assay? Why this bacterium and not isolated

PG from *E. faecalis*? The methods section states a different organism being used (and autoclaved cells). In addition, zymograms are difficult to interpret, since mere PG binding can result in a positive signal. It would be better to back up these data with an actual cell wall degradation assay.

8. Line 131 – can you show an image of what this screen looks like in reality, i.e. a colony with vs. without halo?

9. Line 146 – 154 seems extraneous, almost like a piece of the abstract got somehow transferred here?

10. The luciferase assay in Fig. S5 needs a positive control. Also, why is the data distribution for AtIA and AtladlysM+AdmA so strange (with a very large range)? Also mention what the data points and error bars are in the figure legend.

11. I am not sure I understand how the demographs work with cell chains. For example, the lysM truncation mutants have a chaining phenotype. Are the demographs just depicting the length of one cell, or several cells in a chain? If the former, why does AtIA localization split, as if those were two separate cells? This needs to be explained better. Also, all demographs give distance from mid-cell in “mm”, this should be “ μm ”

12. Western Blots (e.g., Fig. S2BCD) need a loading control. Also, what are bands not marked with an asterisk? Some of these appear to have a positive signal in the zymogram gel (e.g., altA-1-5). Are these activated cleavage product resulting from protease activity?

Please note that all references to page and line numbers refer to the marked copy of the manuscript.

Reviewer #1 (Remarks to the Author):

The manuscript by Salamaga et al reports a moonlighting role for LysM peptidoglycan binding domains. They showed that the LysM domain of the N-acetylglucosaminidase AtIA is essential for the localization at the division site in *Enterococcus faecalis*. Using transposon mutagenesis, they also identify a membrane-bound protein AdmA that is required for the recruitment of AtIA at the septum.

Comments to the Authors:

1. Lines 70-71: The authors say that the *atIA-gfp* fusion had a minor impact on AtIA activity. However, according to the results shown in Fig. 1B, the *atIA1-6-gfp* strain showed a significant increase in cell chain length ($P = 1.42 \times 10^{-6}$). It would be a much better demonstration if the authors show the actual cell length of these strains. Does the GFP fusion change the stability of AtIA?

The very low P is not a proxy for the amplitude of the difference in cell chain length value but rather reflects the consistency of measurements (homogenous size of the population). The point we made was based on the median values for cell chain length measurements: the average values for the *atIA-gfp* strain ($FSC_{log}=76.0 \pm 0.6$) are much closer to the chain size of the parental strain ($FSC_{log}=46.2 \pm 0.8$) than that of the *atIA* mutant ($FSC_{log}=485.42 \pm 47.0$). We have now provided a supplementary figure (Fig. S2A) comparing the number of cells per chain for the WT and *atIA-gfp*. For the $\Delta atIA$ mutant, we cannot provide a reliable number of cells per chain because the chains are so long that most of them are out of focus. We have added a picture showing a field for each strain (Fig. S2B). The text has been modified to include a reference to the number of cells per chain (p.4, l. 76-80). The data provided indicate average cell chain lengths of 2.3 for WT and 3.8 for the *atIA-gfp* strain. We have therefore maintained our conclusion but changed the wording, referring to a limited change in cell chain length rather than a "minor" change.

Western blot analyses indicate that the AtIA-GFP fusion is stable. Anti-GFP antibodies detect a major band that corresponds to the full-length fusion. The less intense band corresponds to a truncation of the N-terminal domain (this band is also detected by anti-AtIA antibodies (Fig. S2C), not to a breakdown of the GFP fusion).

2. Lines 80-82: it is unclear why the authors used AtlB to construct these GFP fusion strains. What's the relationship between AtlA and AtlB? What's the function of AtlB?

AtlB is the only other *E. faecalis* autolysin containing a LysM domain that has been characterized (Mesnage *et al*, 2008, JBC). AtlB is an *N*-acetylmuramidase that can cleave the septum but with much less efficiency than AtlA. We previously demonstrated that swapping the LysM domain of AtlA for the LysM domain of AtlB drastically reduces the septum cleavage activity of recombinant proteins (Mesnage *et al*, 2008, JBC). Based on the data presented showing that AtlA is specifically targeted to the septum (Fig. 1), we hypothesized that the functional specialisation of AtlA may be underpinned by its LysM domain. We therefore decided to compare the contribution of LysM domains from AtlA and AtlB to the subcellular localisation of GFP. We have added information about AtlB to the revised manuscript (p.3, l.46-49).

3. Fig. 1E and 1F, why the GFP protein shows several bands in the first lane?

Recombinant GFP has been shown to multimerize at high concentrations (DOI: 10.1016/j.ab.2007.05.025; doi: 10.1107/S1744309112028667). We have added a sentence to indicate that the first lane corresponds to recombinant GFP as a positive control and a comment to explain the presence of multiple bands (in the legend). See p.15, l.416-418)

4. Line 95: it is unclear why the authors claim that the GFP fluorescence in the SBGMB strain was associated with the "old" peptidoglycan.

Reviewer 3 made a similar comment (comment 5). To avoid any confusions, we replaced "old peptidoglycan" by "poles".

5. Fig. S2: Authors should provide the immunoblotting analysis with culture supernatants. What does the *atlAE212Q* strain mean in the Fig. S2A?

We have repeated the immunoblot experiments and replaced the panels B and C. The conclusions remain unchanged. The data show an accumulation of the pre-proteins in crude extracts for AtlA proteins with a LysM domain of 4 repeats. The intensity of the signal in culture supernatants is decreasing as the number of LysM repeats is reduced.

Fig. S2 has been moved to the main text and is now Fig. 3.

The *altA_{E212Q}* mutant is a JH2-2 derivative with an E→Q-amino acid substitution in position 212 that abolishes AtlA catalytic activity. This mutant has been described in a recent manuscript (Zamboni *et al.*, 2022, JBC). We have described this mutation in the legend of Fig. 3 and added this strain to Table S2 with a reference to the manuscript describing this strain (Zamboni *et al.*, 2022).

6. Line 126: Fig. S3E is missing in the supplementary material.

Reference to Fig. S3E has been removed.

7. Lines 130-138: It would be nice to see the halos of these transposon mutants. Why the authors have chosen AdmA for further investigation, in particular that this is a hypothetical protein.

We have added a supplementary figure showing that the method specifically detects AtlA activity (Fig. S4A). The figure also contains pictures of plates with transposon mutant candidates showing an altered AtlA activity (Fig. S4B) and a direct comparison of the WT and $\Delta admA$ phenotype using this assay (Fig. S4C).

We focused on AdmA for three major reasons: (i) the *admA* mutant has a very clear phenotype; (ii) it is a protein of unknown function and (iii) it is predicted to be intracytoplasmic, making it a perfect candidate to recruit AtlA inside the cell.

We have cited the supplementary figure showing halos (p.6, l.160) and added a comment in the text to explain why we chose *admA* for further analysis (p.6, l.166-167)

8. Lines 151-154: It has not been established that the LysM domain promotes the recruitment of AtIA at the septum via AdmA. Only a correlation is made. In the absence of further evidence, they should tread more lightly here.

The original wording did not infer a direct interaction with AdmA but we have changed the text and removed the reference to AdmA (p.8, l.182).

9. Fig. 3C: the complementation of the $\Delta admA$ should be also performed.

We have provided a figure showing the complementation of the *admA* mutation in the *atla-gfp* background (now Fig. 5B). Because the complementation is partial, we decided to show fields corresponding to the parental and the complemented strains to show more clearly that the expression of *admA* on an inducible plasmid restores the presence of fluorescence foci at the septum and poles.

10. Several figures in the supplementary material should be moved to the main text, such as Fig. S2, Fig. S3.

Done. Fig. S2 is now Fig. 3 and Fig. S3 is Fig. 4.

11. Lines 160-164: the authors indicate that the AtIA protein possesses an ESPR. This information should be included in the introduction section.

We have changed the text as suggested by the reviewer (p.3, l.53-54).

12. The methodology for transposon mutagenesis and zymogram analysis should be much more detailed.

We have described the transposon mutagenesis, screening and insertion mapping (p.11-12, l.286-307). The description of zymograms is also described in detail (p.13, l.341-345)

13. There are also many mistakes along the manuscript. For instance:

1) Line 59: the appropriate reference should be provided.

A reference has been added (p.3, l.63).

2) Line 62: the reference 10 should be removed from this sentence.

Done.

3) Line 68: the authors should clarify the strain *atIA1-6-gfp*.

Done.

4) "localisation" should be "localization".

Typo corrected throughout the manuscript (15 changes).

5) The authors used "JH2-2" in Fig. 1B and Fig. 3, but "WT" in Fig. 1E and 1F.

JH2-2 has been replaced by WT in all figures.

6) Line 111: *M. lysodeikticus* or *Micrococcus luteus*?

The cells purchased from SIGMA are labelled as *M. lysodeikticus* but the correct name is *M. luteus* so the latter should be used. The text has been changed throughout the manuscript to be consistent (5 changes).

7) Line 123: the result for zymogram analysis is not shown in Fig. S3C.

The zymogram has been added (Figure 4C).

8) Fig. S2: What do the arrows mean?

The arrows correspond to N-terminal truncation products. To avoid any confusion, these have been removed.

9) Table S2: the description for each strain or plasmid should be unique.

The description of all strains has been checked and modifications have been made to have unique strain descriptions.

Reviewer #2 (Remarks to the Author):

Review: A moonlighting role for LysM peptidoglycan binding domain underpins daughter cell separation.

This work sought to understand how AtIA is localized to the septum. Salamanga et al. claimed that the LysM domain of AtIA is required for its septal localization. In addition, the authors suggested AtIA is recruited to the septum before secretion (lines 31 and 64). Also, they identified six genes (Table S1) that when inactivated by a transposon, led to a decrease in AtIA function. Among them is AdmA, a transmembrane protein that is seemingly required for AtIA secretion and its septal localization. From these results, the author concluded that the autolytic activity of AtIA is “under the control of several mechanisms that act synergistically” (line 196). This claim, and several others, are nevertheless not substantiated sufficiently by the data presented. Furthermore, several alternative explanations of the results were not fully addressed. While the preliminary findings are interesting, some experiments lack the key controls required to draw definitive conclusions.

The comment l. 196 refers to this work AND our previous study (Salamanga et al., *PLoS Pathogens*), which described that AtIA activity is under the control of posttranslational modifications including the glycosylation of its N-terminal domain (that inhibits activity) and proteolytic cleavage by the extracellular proteases including GelE (that activate the enzyme). The current work shows that the control of its subcellular localization underpins daughter cell separation after cell division. We believe that our previous work (Salamanga et al., *PLoS Pathogens*) together with this study provide a substantial body of experimental data supporting this claim. We have added a reference to our paper p.9, l. 225.

Major concerns

- Fig. 1C to F: This set of experiments attempted to show that the LysM domain of AtIA is sufficient to localize it to the septum. The authors fused the signal peptides of AtIA or AtIB to the N-terminus of GFP. Then, the LysM domains from these two proteins were added to the C-terminus of the GFP fusion. However, the “sAg”, “sAgmB”, and “sBgmB’ fusions are not secreted and thus they could be non-functional or misfolded. These constructs are not interpretable. If I removed them from the analysis, comparing “sAgmA” and “sBgmA” suggests the signal peptide played an important role in the septal localization of AtIA, not the LysM domains.

We disagree with the reviewer. We maintain that the “sAg”, “sAgmB”, and “sBgmB’ fusions can be interpreted. The fact that they are “non-functional” is precisely the reason why they provide meaningful information. Experimental data provided in the manuscript and in our previous work support this idea.

Our western blot experiments show that the fusion proteins are stable (a predominant band corresponding to the full-length fusions is systematically observed in Fig. 1) and our fluorescence microscopy experiments argue against misfolded proteins because the GFP is fluorescent. The septal localization of the sAgmA fusion (demographs) and translocation to the cell surface (western blots) indicate that the sAg part of the fusion is functional. In a previous study (Mesnage et al., 2008), we overexpressed and purified a recombinant variant of AtIA (AtIAB). In AtIAB the LysM domain of AtIA is swapped for the LysM domain of AtIB; this protein is **produced at high levels in the *E. coli* cytoplasm; it is stable and active**, therefore supporting the idea that the fusions containing the AtIB LysM domain described in this (built according to the same strategy) can be interpreted. Based on the gain of function provided by the addition of LysMA to the sAg fusion (sAgmA strain), we maintain our conclusion that the LysM domain of AtIA is critical for septal localization. The comparison with the sAgmB fusion indicates that the LysM domain also plays a critical role for septal localisation. This conclusion is supported by the rest of our data.

- It is unclear why AtIB is relevant, except they are both autolysins and contain LysM repeats. There is not sufficient background about AtIB to justify why it is included in the study.

More information is now provided in the introduction. See response to reviewer 1 and revised manuscript (p. 3, l. 46-50).

-Fig. 2: Are these constructs secreted? Is there any evidence that shows that the constructs are not degraded and produced at similar levels?

We have performed immunoblotting experiments using anti-GFP antibodies and included this data in Fig. 2B. All proteins are secreted and stable. As shown for the WB in Fig. 3, a band corresponding to the pre-protein is detected in cell lysates of strains with a truncated LysM domain, indicating that the truncation of the LysM domain impairs secretion.

- If AtIA is not secreted (Fig. 3B), the cells should fully phenocopy $\Delta atIA$ (Fig. 3A), but they are not.

The *admA* mutation still allows AtIA secretion and surface display (Fig. 4C). We are therefore not expecting that the deletion of the gene would phenocopy the $\Delta atIA$ mutant.

Also, there is no complementation for Fig. 3C. And the link between AdmA and the LysM repeats, if any, is not substantiated.

We have now added a figure showing the complementation of the *admA* mutation (see reviewer 1, point 9).

We agree with the reviewer that we have not been able to provide evidence showing a direct interaction between AdmA and AtIA LysM domain. This limitation of our work is clearly stated in the revised manuscript (p.9, L. 218-221).

Is there any evidence indicating the loss of AtIA is the reason why the $\Delta admA$ mutant forms chains?

We do not infer that AtIA is **lost**; our data indicates that AtIA is still produced but sequestered in the cytoplasm in the absence of AdmA (as evidenced by a band corresponding to the pre-protein). We have provided evidence in the revised manuscript that in the *admA* mutant, the level of surface-associated AtIA is reduced (Fig. S4). Our conclusion is based on compelling evidence in the literature that AtIA is the major autolysin responsible for cell separation septum (Qin et al., 1997 Antimicrobial Agent Chemother, doi:

10.1128/AAC.42.11.2883; Mesnage et al., 2008, J Biol Chem, doi: 10.1074/jbc.M802323200).

-The authors mentioned truncating the LysM repeats reduced the amount of surface displayed AtIA (Fig. S2A), but such truncation did not affect its localization (lines 102-104). This result contradicts their major conclusion that LysM is required to recruit AtIA to the septum.

The key conclusion from Fig. S2 (now Fig. 3) is that the amount of protein detected at the cell surface by immunofluorescence decreases when the LysM domain of AtIA is truncated. Given the low intensity of the signal for several strains (pointed out by reviewer 3), we decided to remove the original statement that appeared contradictory (“Interestingly, it did not alter the localization of this protein at the septum and poles”).

In Fig. S2B, it is unclear why whole cultures were loaded rather than cell-free supernatant. For some reason, the student who performed the immunoblotting did not use supernatants so we did not provide this data in the original submission. Even though the amount of protein in supernatants could be inferred from the comparison between whole cultures (cells + supernatants) and cell lysates, we decided to repeat the experiments to provide consistent analyses throughout the manuscript. The data presented in the new figure (now Fig. 3B and 3C) is entirely consistent with the original one and do not change the conclusions drawn.

The speculation that the top band is the preprotein of AtIA, which is crucial to their claim, has not been fully tested.

Several facts strongly support the hypothesis that this band is the preprotein of AtIA:

(i) this band has a molecular weight higher than the full length, secreted AtIA and compatible with that of the pre-protein.

(ii) it is only found in the strains producing AtIA variants that are sequestered in the cytoplasm (as demonstrated by immunofluorescence).

(iii) it is recognised by anti-AtIA antibodies and has peptidoglycan hydrolytic activity.

We cannot think about an alternative hypothesis but if the reviewer has one, we would be happy to include it in the text.

-Fig. S3D confirmed my concern that “sAgmB” in Fig. 1D is not interpretable. Unlike “sAgmB”, the full-length AtIA protein with the AtIB LysM domain is septal localized. Is AtIB targeted to the septum?

The strain *atIA*_{1-6HB} analysed in Fig. S3D (now Fig. 4) produces an AtIA variant with 6 LysM repeats from the *E. faecalis* peptidoglycan hydrolase AtIB. Individual LysM repeats from AtIA (LysMA1 to LysMA6) were alternatively substituted by one of the two AtIB LysM repeats (LysMB1 and LysMB2), keeping the linkers between repeats identical to those found in AtIA (see diagram representation in Fig. 4A). Fig. 4D shows that the LysM_{1-6HB} domain can provide septal localization (albeit not strictly restricted to the septum), thereby supporting the idea that the cytoplasmic localization of “sAgmB” in Fig. 1D can be interpreted.

Despite several attempts, we have never been able to establish the subcellular localization of AtIB, most likely due to the low expression level of this protein (Salamaga *et al.*, 2017).

Neither *gfp* fusions nor immunofluorescence experiments led to conclusive experiments.

-Fig. S5C, why AtIA alone and AtIA Δ lysM +AdmA generate luminescence?

We are unable to explain this unexpected result and given the variability of independent experiments, we decided to not draw any conclusion from these experiments.

-Line 94-95: what is the evidence that suggests “sBgmB” is associated with old peptidoglycan? Protein aggregates can often be directed to the cell pole as well. See response to reviewer 1, comment 4.

-The Materials and Methods section is unacceptably brief.

There is no description of the immunofluorescent microscopy and the preparation of the whole cell.

A paragraph has been added to describe immunofluorescent microscopy (p.13, l.359-365). *E. faecalis* cell extracts preparation has been modified (p.13, l.324-326).

Also, the transposon mutagenesis and insertion mapping had a single-line description. We have now provided a detailed description of the mutagenesis, mutant screening, and insertion mapping (also described in detail in Smith *et al.*, 2019, Figure S10) (p. 11-12, l. 286-307)

What is the strain used to construct the transposon mutants? How was the screen done?

The strain used to build the mutant library was OG1RF and *admA* was originally identified in this strain. Our preliminary experiments confirmed that the in-frame deletion of *admA* in OG1RF leads to a reduced AtIA peptidoglycan hydrolase activity (see panel A); zymogram analysis also revealed an accumulation of the AtIA pre-protein in the OG1RF *admA* mutant (see black arrow in panel B).

Characterization of the *admA* in-frame deletion mutant in the OG1RF genetic background. A. AtIA hydrolytic activity on BHI-agar supplemented with *M. lysodeikticus* autoclaved cells. The *admA* mutant displays a reduced hydrolytic activity as compared to the parental strain whilst the *atIA* mutant shows no activity at all. B. Cell-associated and supernatant activities of AtIA. Peptidoglycan hydrolytic activity was detected by zymogram using *M. lysodeikticus* as a substrate. Black arrow, preprotein; white arrow, secreted protein; grey arrows, N-terminal and LysM truncated AtIA.

To be consistent with our previous work that was carried out in the JH2-2 genetic background (Salamaga *et al.*, 2017; mesnage *et al.*, 2008), strain JH2-2 was chosen to further analyse AdmA function. The *admA* in-frame deletion mutant in the OG1RF background displays the same phenotype as its counterpart in JH2-2 (see figure). The screen has now been described in detail (p. 11,-12 L. 293-297); see also response to reviewer 1, point 7).

How many colonies were screened?

Approximately 20,000 mutants were screened. This information has been added to the revised manuscript (p.7, l.161).

The ITC and split luciferase assays were not described sufficiently. How were the proteins purified? ITC should be Fig. S4, not S5.

We have added information in the material and methods to describe the purification recombinant proteins used and briefly described the AdmA and LysM purifications (p.14, l.368-380). We also added a gel showing the 2 protein purification products after gel filtration in the same buffer (to avoid dilution heat) that were used for ITC experiments. All the information for ITC is in the legend of the corresponding figure.

The method used for luciferase assay is exactly the one described in the paper we cite. Table S2 and the material and methods section describing all strains, plasmids and oligonucleotides already contains a substantial amount of information. Since the experiment provided a negative result, we do not wish to add all the information about the split luciferase material in the manuscript. In agreement with the data sharing policy of the journal, we are

happy to share any specific information or material upon request. All plasmids have been fully sequenced (Plasmidsaurus).

Minor comments:

-Figure 1E and 1F: Is lane 3 the protein ladder?

Correct. This lane has been labelled in the revised figure (MWM).

What is GFP? Is it recombinant GFP loaded as a positive control?

Green Fluorescent Protein. We loaded recombinant GFP as a positive control. The excess of protein led to a multimerization of the protein (DOI: 10.1016/j.ab.2007.05.025; doi: 10.1107/S1744309112028667). See reviewer 1 comment 3, we added a comment in the figure legend.

-Line 101: The immunofluorescent microscopy of AtIA was used to show that it is surface-exposed. What is the basis of this?

Immunofluorescence will only detect proteins that are accessible to antibodies at the cell surface.

The loss of signal in the LysM mutants can be caused by the inability to bind peptidoglycan, not necessarily means that AtIA is not surface exposed.

Correct. The point we make here is that AtIA requires a full LysM domain to be exposed at the cell surface.

-Line 103: The signals in Fig S2A are weak (atIA1-3, atIA1-2, and atIA1). How could the authors conclude the localization of these proteins is not altered?

This comment is related to a previous one about lines 102-104 (see above). We have addressed the reviewer's comment and deleted the sentence.

-Figure S2A: Is atIAE212Q a catalytic site mutant? It is not mentioned in the text. Where is E212?

Correct. AtIA_{E212Q} is described in Table S2. See response to reviewer 1, point 5.

-Figure S2 legends: Typo of (E), should be (D).

Changed

-There is no evidence of AdmA and AtIA colocalizing. Thus, the title of Figure 3 should be changed.

The title has been changed to "AdmA is preferentially localised at the septum and is essential for AtIA recruitment at the septum"

-What is the correlation between FSC and the chain length?

The point of the flow cytometry experiment is to compare the cell chain length of the *admA* mutant with the parental strain and demonstrate that the complementation restores (at least partially) the defect caused by the *admA* mutation. As previously mentioned, it is technically challenging/impossible to count the number of cells per chain when strains form long chains (such as in the *atIA* mutant, LysM truncations or with the *admA* mutant). This is the reason why we do not systematically attempt to correlate FSC and chain length.

Monitoring forward scattered light as a proxy for chain length is well established in the literature. We also argue that the correspondence between FSC and cell chain length is an information that would not alter our conclusions.

-Line 273: Please specify what is TCA.

TCA has been replaced by trichloroacetic acid.

Reviewer #3 (Remarks to the Author):

The manuscript by Salamaga et al. describes a novel function for LysM domains (typically associated with peptidoglycan binding) as an intracellular septal recruitment signal, likely by the newly-identified AdmA protein. This is pretty straight-forward work, but I have substantial comments on data organization and presentation.

1. Please keep *E. faecalis* in the title. This does not seem to be a general, well-conserved function, but rather specific to *E. faecalis*.

Done. We have added *E. faecalis* in the title.

It is worth mentioning that this function is not specific to *E. faecalis*. *Lactococcus lactis* encodes an *admA* ortholog (*lmg_0639*) with a similar function. The characterisation of the *L. lactis lmg_0639* has been reported at the 11th Symposium on Lactic Acid Bacteria 2014 but this work has not been published in a peer-reviewed journal yet.

2. The introduction could use some more information on AtlA biochemistry. How does a glucosaminidase contribute to septal cleavage?

We struggled to understand what the reviewer meant. AtlA glucosaminidase activity has been previously characterized (Eckert et al., 2006) and determining “how AtlA contribute to septal cleavage” is the aim of this study.

3. Fig. 1A: Is *atlA1-6* the same as wt AtlA? This is not clear. In Fig. 1C, please add labels to the color to make it more clear to the reader.

Correct, *AtlA1-6* is the same as wt AtlA. We have replaced *atlA₁₋₆-gfp* by *atlA-gfp*.

4. The entire description of the truncated constructs (line ~81 – 95) is very difficult to follow and should be streamlined. Please always mention the identity of the constructs (what is sAgmB stand for? The reader has to go back and forth between figure legend and the text to make sense of it).

We have explained the rationale behind the name of each construct and referred twice to figure 1C that represents a schematic diagram of each fusion (p.4, l.86-90).

5. Line 95: There is no evidence of old peptidoglycan. Perhaps change to “pole”

Done.

6. Line 99 states “removing LysM...abolished restricted localization...” and in line 103 “interestingly, it did not alter the localization of this protein to the septum”. This seems like a contradiction. Also, in Fig. S2A, there is no visible fluorescence in half of the images, so this statement is difficult to evaluate.

Reviewer 2 made the same comment (see minor comments). We have removed this statement in the revised manuscript as it was not the major point we wanted to make.

7. Zymogram analysis: line 111 states that “*M. lysodeikticus*” was used. This is unclear. Was PG from that organism isolated? Did they use a whole cell in the assay? Why this bacterium and not isolated PG from *E. faecalis*? The methods section states a different organism being used (and autoclaved cells). In addition, zymograms are difficult to interpret, since mere PG binding can result in a positive signal. It would be better to back up these data with an actual cell wall degradation assay.

The substrate used for zymogram and agar plate assays consists of autoclaved *M. luteus* cells. This is described in the Material and methods (p.13, l. 341-345). As shown in the new supplementary Fig. S4, AtlA is the only peptidoglycan hydrolase in *E. faecalis* able to

solubilise this substrate in zymograms and on agar plate assays. The reason for this is unknown. We therefore used *M. luteus* autoclaved cells as a substrate to specifically detect AtIA activity. Freeze-dried cells were purchased from SIGMA (product ref M3770). We agree with the reviewer that zymograms can provide false-positive results, but this has only been described with very high amounts of pure proteins (several micrograms). We are confident that the very low amount of AtIA protein in crude extracts and supernatants (tens of ng at most) is unlikely to be an artifact. Zymogram analysis using cell extracts from mutants described in our recent study (Zamboni et al., J Biol Chem, 2022) shows that no autolytic band is detected in catalytic mutants (see below). All proteins are stable (Fig. 6 in Zamboni et al), so this is the demonstration that the bands observed on a zymogram results from peptidoglycan hydrolysis rather than an artefact.

It is also worth mentioning that *E. faecalis* encodes 12 proteins with LysM domains. If the presence of a clearing zone was an artifact, then we would expect to detect many bands by zymograms. This is not the case.

8. Line 131 – can you show an image of what this screen looks like in reality, i.e. a colony with vs. without halo?

The information requested by the reviewer is shown in the new Fig. S4B.

9. Line 146 – 154 seems extraneous, almost like a piece of the abstract got somehow transferred here?

We agree with the reviewer. The manuscript has been transferred from one journal to another without being reformatted. The section mentioned by the reviewer is the start of the discussion. In response to several reviewers' comments, we have now added headings to help the readers to follow the narrative and extra information in the introduction, results, material and methods and discussion. Please note that the Discussion is now a separate section.

10. The luciferase assay in Fig. S5 needs a positive control. Also, why is the data distribution for AtIA and AtIA Δ lysM+AdmA so strange (with a very large range)? Also mention what the data points and error bars are in the figure legend.

We have plasmid 1 encoding the full-length luciferase as a positive control. The luciferase assays have been repeated many times (our last 3 attempts are presented) and as pointed out by the reviewer, the data simply don't make sense! Not only there is a huge variability in the activity detected, but the few assays with strong activity do not make sense because they are not consistent with their internal controls. For example, there is some activity with the AtIA Δ lysM+AdmA assay but far less with the AtIA+AdmA assay. We felt like readers would like to be aware about the experiment we tried, even though we could not conclude much from these experiments. All plasmids were fully sequenced (Plasmidsaurus) so we cannot explain why this assay did not work. To the best of our knowledge, the split luciferase assay has not been attempted with *E. faecalis*.

11. I am not sure I understand how the demographs work with cell chains. For example, the lysM truncation mutants have a chaining phenotype. Are the demographs just depicting the length of one cell, or several cells in a chain? If the former, why does AtIA localization split, as if those were two separate cells? This needs to be explained better. Also, all demographs give distance from mid-cell in "mm", this should be " μ m"

Demographs depict the length of one cell. The "split" in AtIA localization reflects the fact that it can be found on equatorial rings, which correspond to the new site of cell division. mm has been changed to μ m

12. Western Blots (e.g., Fig. S2BCD) need a loading control. Also, what are bands not marked with an asterisk? Some of these appear to have a positive signal in the zymogram gel (e.g., altA-1-5). Are these activated cleavage product resulting from protease activity?

We have added a loading control in all figures showing immunoblotting.

As stated in the legend, the bands marked with an asterisk correspond to unprocessed AtIA (preprotein). They cannot be "activated cleavage products resulting from protease activity" because their molecular weight is larger than the molecular weight of the full length secreted AtIA protein produced by the WT strain.

Reviewers' comments:

Reviewer #1 (Remarks to the Author):

All my concerns have been addressed.

Reviewer #2 (Remarks to the Author):

The revised manuscript is significantly improved. Yet, there are a few major concerns that the authors should address before publication.

Authors:

The comment l. 196 refers to this work AND our previous study (Salamaga et al., PLoS Pathogens), which described that AtIA activity is under the control of posttranslational modifications including the glycosylation of its N-terminal domain (that inhibits activity) and proteolytic cleavage by the extracellular proteases including GelE (that activate the enzyme). The current work shows that the control of its subcellular localization underpins daughter cell separation after cell division. We believe that our previous work (Salamaga et al., PLoS Pathogens) together with this study provide a substantial body of experimental data supporting this claim. We have added a reference to our paper p.9, l. 225.

Reviewer:

l. 225. The information about N-terminal glycosylation and GelE-mediated proteolysis should be included rather than just listing the reference.

l.87 Change "discretion" to "secretion"

l.116 Change "cultiure" to "culture"

Reviewer:

- Fig. 1C to F: This set of experiments attempted to show that the LysM domain of AtIA is sufficient to localize it to the septum. The authors fused the signal peptides of AtIA or AtIB to the N-terminus of GFP. Then, the LysM domains from these two proteins were added to the C-terminus of the GFP fusion. However, the "sAg", "sAgmB", and "sBgmB' fusions are not secreted and thus they could be non-functional or misfolded. These constructs are not interpretable. If I removed them from the analysis, comparing "sAgmA" and "sBgmA" suggests the signal peptide played an important role in the septal localization of AtIA, not the LysM domains.

Authors:

We disagree with the reviewer. We maintain that the "sAg", "sAgmB", and "sBgmB' fusions can be interpreted. The fact that they are "non-functional" is precisely the reason why they provide meaningful information. Experimental data provided in the manuscript and in our previous work support this idea.

Reviewer:

There are many reasons why a fusion protein is non-functional. Without eliminating alternative explanations, one cannot conclude that "the signal peptide and the LysM domain are both required for septal localization.". For example, some of the GFP fusions could be aggregating and were not secreted by the SEC system. As a result, these proteins are not even in the same compartment. The LysM domain may prevent the GFP from aggregating instead of being required for secretion.

In other words, how do you know if the difference between the phenotype of sAg and sAgmA is not an artifact due to GFP aggregation?

I would like the authors to reconsider reviewer 3's suggestion (point 4) to clarify the naming system of the constructs. It is frustrating to go back and forth between Fig. 1C and the text to understand these constructs.

Authors:

Our western blot experiments show that the fusion proteins are stable (a predominant band corresponding to the full-length fusions is systematically observed in Fig. 1) and our fluorescence microscopy experiments argue against misfolded proteins because the GFP is fluorescent. The septal localization of the sAgmA fusion (demographs) and translocation to the cell surface (western blots) indicate that the sAg part of the fusion is functional.

Reviewer:

The GFP fusion is fluorescent does not mean that it is not aggregated. Cautions must be made to avoid misinterpretation of protein fusions. I am not arguing sAgmA is not functional, but it cannot be used to make the inference that "the sAg part of the fusion is functional" and therefore sAgMB is functional. sAgMB is not even secreted, indicating the sAg part cannot be functional. The authors argue that it is because the LysM domain of AtIA is missing. But it could be sAgmA is not aggregating but the other constructs are.

Authors:

In a previous study (Mesnage et al., 2008), we overexpressed and purified a recombinant variant of AtIA (AtIAB). In AtIAB the LysM domain of AtIA is swapped for the LysM domain of AtIB; this protein is produced at high levels in the E. coli cytoplasm; it is stable and active, therefore supporting the idea that the fusions containing the AtIB LysM domain described in this (built according to the same strategy) can be interpreted.

Reviewer:

Again, one cannot use another fusion construct to infer the functions of sAg and sAgmA. Being able to swap LysM domains of AtIA and AtIB in E. coli does not mean that fusing the LysM domain and the signal peptide with GFP in E. faecalis will generate functional proteins.

Additional controls will be needed to be done to justify the effect is due to the LysM domain instead of other trivial reasons like protein aggregation. For example, the authors should consider using monomeric variants like mNEONGreen. The fact that these GFP fusions are not secreted is alarming, to say to least.

Authors:

Based on the gain of function provided by the addition of LysMA to the sAg fusion (sAgmA strain), we maintain our conclusion that the LysM domain of AtIA is critical for septal localization. The comparison with the sAgMB fusion indicates that the LysM domain also plays a critical role for septal localisation. This conclusion is supported by the rest of our data.

Reviewer:

sAgMB is not secreted. The conclusion is not definitive as the alternative hypothesis is still viable. The authors should at least tone down their statements to acknowledge other possibilities.

Reviewer:

The speculation that the top band is the preprotein of AtIA, which is crucial to their claim, has not been fully tested.

Authors:

Several facts strongly support the hypothesis that this band is the preprotein of AtIA:

(i) this band has a molecular weight higher than the full length, secreted AtIA and compatible with that of the pre-protein.

(ii) it is only found in the strains producing AtIA variants that are sequestered in the cytoplasm (as demonstrated by immunofluorescence).

(iii) it is recognised by anti-AtIA antibodies and has peptidoglycan hydrolytic activity.

We cannot think about an alternative hypothesis but if the reviewer has one, we would be happy to include it in the text.

Reviewer:

With Fig. 3B and 3C, the reviewer found the arguments sufficient. These statements should be included in the text.

Reviewer:

-Fig. S3D confirmed my concern that "sAgmB" in Fig. 1D is not interpretable. Unlike "sAgmB", the full-length AtIA protein with the AtIB LysM domain is septal localized. Is AtIB targeted to the septum?

Author:

The strain atIA1-6HB analysed in Fig. S3D (now Fig. 4) produces an AtIA variant with 6 LysM repeats from the *E. faecalis* peptidoglycan hydrolase AtIB. Individual LysM repeats from AtIA (LysMA1 to LysMA6) were alternatively substituted by one of the two AtIB LysM repeats (LysMB1 and LysMB2), keeping the linkers between repeats identical to those found in AtIA (see diagram representation in Fig. 4A). Fig. 4D shows that the LysM1-6HB domain can provide septal localization (albeit not strictly restricted to the septum), thereby supporting the idea that the cytoplasmic localization of "sAgmB" in Fig. 1D can be interpreted.

Reviewer:

The AtIA1-6HB cannot be used to justify sAgmB. They are very different constructs. AtIA1-6HB has 3x more of the AtIB LysM domains. It has no GFP and with the catalytic domain of AtIA. Importantly, sAgmB does NOT localize to the septum, despite having "sA" and "mB". I am suggesting the problem is the "g".

Authors:

The ITC and split luciferase assays were not described sufficiently. How were the proteins purified? ITC should be Fig. S4, not S5.

We have added information in the material and methods to describe the purification recombinant proteins used and briefly described the AdmA and LysM purifications (p.14, l.368-380). We also added a gel showing the 2 protein purification products after gel filtration in the same buffer (to avoid dilution heat) that were used for ITC experiments. All the information for ITC is in the legend of the corresponding figure. The method used for luciferase assay is exactly the one described in the paper we cite. Table S2 and the material and methods section describing all strains, plasmids and oligonucleotides already contains a substantial amount of information. Since the experiment provided a negative result, we do not wish to add all the information about the split luciferase material in the manuscript. In agreement with the data sharing policy of the journal, we are happy to share any specific information or material upon request. All plasmids have been fully sequenced (Plasmidsaurus).

Reviewer:

I disagree. The submission guideline clearly stated that "authors must ensure that their Methods section includes adequate experimental and characterization data necessary for others in the field to reproduce their work.". Besides reproducibility, the readers cannot interpret why the result is negative without sufficient experimental details provided.

Reviewer #3 (Remarks to the Author):

The manuscript has much improved and I have no further major concerns.

Please just address these additional minor concerns that weren't addressed in revision (It looks like I was not very clear in my critique):

1. "2. The introduction could use some more information on AtIA biochemistry. How does a glucosaminidase contribute to septal cleavage?"

Author response: We struggled to understand what the reviewer meant. AtIA glucosaminidase activity has been previously characterized (Eckert et al., 2006) and determining "how AtIA contribute to septal cleavage" is the aim of this study."

My response: Upon re-reading I realize my comment was not very clear. What I meant was that the manuscript would benefit from a bit of discussion about glucosaminidases and how they may contribute to septal cleavage in general (so some discussion of septal PG architecture, more detail on the cleavage specificity of a glucosaminidase). This would help the reader put everything into context, and especially help readers who are not familiar with cell wall biochemistry. In addition, it would help people with a Gram-negative background, who typically think of amidases as septal cleavage enzymes.

2. Western Blots (e.g., Fig. S2BCD) need a loading control. Also, what are bands not marked with an asterisk? Some of these appear to have a positive signal in the zymogram gel (e.g., altA-1-5). Are these activated cleavage product resulting from protease activity?"

Author response: We have added a loading control in all figures showing immunoblotting. As stated in the legend, the bands marked with an asterisk correspond to unprocessed AtIA (preprotein). They cannot be "activated cleavage products resulting from protease activity" because their molecular weight is larger than the molecular weight of the full length secreted AtIA protein produced by the WT strain.

My response: I think the authors missed the word "not" in my comment – I was wondering about the bands that were NOT marked with an asterisk. This could be fixed with a simple arrow indicating "non-specific bands" or the likes, but my point stands that some of these might be degradation products.

Comments from reviewers are in black.

Our original reply in blue

The latest response to reviewers' comments is in red.

Reviewer #1 (Remarks to the Author):

All my concerns have been addressed.

We are delighted to hear that we addressed all the concerns of the reviewer.

Reviewer #2 (Remarks to the Author):

The revised manuscript is significantly improved. Yet, there are a few major concerns that the authors should address before publication.

The comment l. 196 refers to this work AND our previous study (Salamaga et al., PLoS Pathogens), which described that AtIA activity is under the control of posttranslational modifications including the glycosylation of its N-terminal domain (that inhibits activity) and proteolytic cleavage by the extracellular proteases including GelE (that activate the enzyme). The current work shows that the control of its subcellular localization underpins daughter cell separation after cell division. We believe that our previous work (Salamaga et al., PLoS Pathogens) together with this study provide a substantial body of experimental data supporting this claim. We have added a reference to our paper p.9, l. 225.

l. 225. The information about N-terminal glycosylation and GelE-mediated proteolysis should be included rather than just listing the reference.

Done (p.8, l. 218-220)

l.87 Change “discretion” to “secretion”

Done (p.4, l. 90)

l.116 Change “cultiure” to “culture”

Done (p.5, l. 119)

- Fig. 1C to F: This set of experiments attempted to show that the LysM domain of AtIA is sufficient to localize it to the septum. The authors fused the signal peptides of AtIA or AtIB to the N-terminus of GFP. Then, the LysM domains from these two proteins were added to the C-terminus of the GFP fusion. However, the “sAg”, “sAgmB”, and “sBgmB” fusions are not secreted and thus they could be non-functional or misfolded. These constructs are not interpretable. If I removed them from the analysis, comparing “sAgmA” and “sBgmA” suggests the signal peptide played an important role in the septal localization of AtIA, not the LysM domains.

We disagree with the reviewer. We maintain that the “sAg”, “sAgmB”, and “sBgmB” fusions can be interpreted. The fact that they are “non-functional” is precisely the reason why they provide meaningful information. Experimental data provided in the manuscript and in our previous work support this idea.

There are many reasons why a fusion protein is non-functional. Without eliminating alternative explanations, one cannot conclude that “*the signal peptide and the LysM domain are both required for septal localization.*”. For example, some of the GFP fusions could be aggregating and were not secreted by the SEC system. As a result, these proteins are not even in the same compartment. The LysM domain may prevent the GFP from aggregating instead of being required for secretion.

In other words, how do you know if the difference between the phenotype of sAg and sAgmA is not an artifact due to GFP aggregation?

We still argue that “*the signal peptide and the LysM domain are both required for septal localization*” simply because this is what the experimental data show in Figure 1. This statement is therefore factually correct. We added the words “of GFP fluorescence” to be accurate.

Whilst it seems difficult to argue against the reviewer’s suggestion about aggregation (it is virtually impossible to prove it wrong), it seems strange to hypothesize that the GFP would be aggregated in the sAg fusion whilst this is not the case when the LysM domain is added to its C-terminal (after a 10 amino-acid linker). Although this is theoretically possible, the chaperone role of the LysM domain seems unlikely.

It is worth mentioning that the GFP is functional and not aggregated when fused to the signal peptide of AtIB (as expected, it is entirely found in supernatants). This demonstrate that the GFP does not require the AtIA LysM domain to prevent aggregation. We feel that pairwise comparisons and the complementary experiments describing LysM truncations are in favour of our interpretation. However, and as stated above, the reviewer’s point is an alternative explanation that we are willing to include in the revised manuscript. We have added a comment in the revised manuscript to highlight the alternative explanation suggested (p.7, l.189-192).

I would like the authors to reconsider reviewer 3’s suggestion (point 4) to clarify the naming system of the constructs. It is frustrating to go back and forth between Fig. 1C and the text to understand these constructs.

We are struggling to understand how we could clarify the naming system for the following reasons:

- 1) It follows a very simple logic clearly described in the revised version (p.4, l.80-82; S=Signal peptide, g=GFP, MA=LysM from AtIA and MB=LysM from AtIB).
- 2) there is a graphic representation of the constructs in Figure 1.
- 3) Reviewer 3 who made this request was satisfied with the text modifications.

Whilst we cannot think about a simpler and/or more rational way of describing our strains, we would be happy to modify the manuscript using a suggestion made by the reviewer.

Our western blot experiments show that the fusion proteins are stable (a predominant band corresponding to the full-length fusions is systematically observed in Fig. 1) and our fluorescence microscopy experiments argue against misfolded proteins because the GFP is fluorescent. The septal localization of the sAgmA fusion (demographs) and translocation to the cell surface (western blots) indicate that the sAg part of the fusion is functional.

The GFP fusion is fluorescent does not mean that it is not aggregated. Cautions must be made to avoid misinterpretation of protein fusions. I am not arguing sAgmA is not functional, but it cannot be used to make the inference that “the sAg part of the fusion is functional” and therefore sAgMB is functional. sAgMB is not even secreted, indicating the sAg part cannot be functional. The authors argue that it is because the LysM domain of AtIA is missing. But it could be sAgmA is not aggregating but the other constructs are.

This comment is again related to an alternative interpretation of the reviewer that is plausible but, in our opinion, unlikely. We have now included this alternative interpretation of the data in the revised manuscript (p.7, l.189-192) so we feel that we have addressed the concern of the reviewer. The key results from our *gfp* fusions are that (i) you need both the signal peptide of AtIA AND its LysM domain to target the GFP to the septum (Fig. 1D and (ii) the LysM domain is essential for septal targeting (Fig. 2 and 3).

In a previous study (Mesnage et al., 2008), we overexpressed and purified a recombinant variant of AtIA (AtIAB). In AtIAB the LysM domain of AtIA is swapped for the LysM domain of AtIB; this protein is produced at high levels in the *E. coli* cytoplasm; it is stable and active, therefore supporting the idea that the fusions containing the AtIB LysM domain described in this (built according to the same strategy) can be interpreted.

Again, one cannot use another fusion construct to infer the functions of sAg and sAgmA. Being able to swap LysM domains of AtIA and AtIB in *E. coli* does not mean that fusing the LysM domain and the signal peptide with GFP in *E. faecalis* will generate functional proteins. Additional controls will be needed to be done to justify the effect is due to the LysM domain instead of other trivial reasons like protein aggregation. For example, the authors should consider using monomeric variants like mNEONGreen. The fact that these GFP fusions are not secreted is alarming, to say to least.

Yet again, this comment is related to the previous ones and refers to the alternative interpretation proposed by the reviewer. As previously stated, we took this hypothesis into account and the revised version of the manuscript indicates that the retention of the GFP could be associated with protein aggregation that no longer exists in the presence of the LysM domain of AtIA (p.7, l.189-192).

Based on the gain of function provided by the addition of LysMA to the sAg fusion (sAgmA strain), we maintain our conclusion that the LysM domain of AtIA is critical for septal localization. The comparison with the sAgmB fusion indicates that the LysM domain also plays a critical role for septal localisation. This conclusion is supported by the rest of our data.

sAgmB is not secreted. The conclusion is not definitive as the alternative hypothesis is still viable. The authors should at least tone down their statements to acknowledge other possibilities.

See previous comments. We have included the alternative hypothesis made by the reviewer in the revised manuscript (p.7, l.189-192).

The speculation that the top band is the preprotein of AtIA, which is crucial to their claim, has not been fully tested.

Several facts strongly support the hypothesis that this band is the preprotein of AtIA:

(i) this band has a molecular weight higher than the full length, secreted AtIA and compatible with that of the pre-protein.

(ii) it is only found in the strains producing AtIA variants that are sequestered in the cytoplasm (as demonstrated by immunofluorescence).

(iii) it is recognised by anti-AtIA antibodies and has peptidoglycan hydrolytic activity.

We cannot think about an alternative hypothesis but if the reviewer has one, we would be happy to include it in the text.

With Fig. 3B and 3C, the reviewer found the arguments sufficient. These statements should be included in the text.

Done (p. 5, l.120-122).

-Fig. S3D confirmed my concern that “sAgmB” in Fig. 1D is not interpretable. Unlike “sAgmB”, the full-length AtIA protein with the AtIB LysM domain is septal localized. Is AtIB targeted to the septum?

The strain atIA1-6HB analysed in Fig. S3D (now Fig. 4) produces an AtIA variant with 6 LysM repeats from the *E. faecalis* peptidoglycan hydrolase AtIB. Individual LysM repeats from AtIA (LysMA1 to LysMA6) were alternatively substituted by one of the two AtIB LysM

repeats (LysMB1 and LysMB2), keeping the linkers between repeats identical to those found in AtIA (see diagram representation in Fig. 4A). Fig. 4D shows that the LysM1-6HB domain can provide septal localization (albeit not strictly restricted to the septum), thereby supporting the idea that the cytoplasmic localization of “sAgmB” in Fig. 1D can be interpreted.

The AtIA1-6HB cannot be used to justify sAgmB. They are very different constructs. AtIA1-6HB has 3x more of the AltB LysM domains. It has no GFP and with the catalytic domain of AtIA. Importantly, sAgmB does NOT localize to the septum, despite having “sA” and “mB”. I am suggesting the problem is the “g”.

This reviewer's comment has been addressed by highlighting the possibility of GFP aggregation (p.7, l.189-192).

The ITC and split luciferase assays were not described sufficiently. How were the proteins purified? ITC should be Fig. S4, not S5.

We have added information in the material and methods to describe the purification recombinant proteins used and briefly described the AdmA and LysM purifications (p.14, l.368-380). We also added a gel showing the 2 protein purification products after gel filtration in the same buffer (to avoid dilution heat) that were used for ITC experiments. All the information for ITC is in the legend of the corresponding figure. The method used for luciferase assay is exactly the one described in the paper we cite. Table S2 and the material and methods section describing all strains, plasmids and oligonucleotides already contains a substantial amount of information. Since the experiment provided a negative result, we do not wish to add all the information about the split luciferase material in the manuscript. In agreement with the data sharing policy of the journal, we are happy to share any specific information or material upon request. All plasmids have been fully sequenced (Plasmidsaurus).

I disagree. The submission guideline clearly stated that “authors must ensure that their Methods section includes adequate experimental and characterization data necessary for others in the field to reproduce their work.”. Besides reproducibility, the readers cannot interpret why the result is negative without sufficient experimental details provided.

We have added a paragraph in the Methods section to provide details about the ITC (p. 14, l.372-376) and luciferase assays (p. 13, l.379-384). This section now contains all the information required for others to replicate all the experiments described in the manuscript.

Reviewer #3 (Remarks to the Author):

The manuscript has much improved and I have no further major concerns. Please just address these additional minor concerns that weren't addressed in revision (It looks like I was not very clear in my critique):

1. "The introduction could use some more information on AtlA biochemistry. How does a glucosaminidase contribute to septal cleavage?"

We struggled to understand what the reviewer meant. AtlA glucosaminidase activity has been previously characterized (Eckert et al., 2006) and determining "how AtlA contribute to septal cleavage" is the aim of this study."

Upon re-reading I realize my comment was not very clear. What I meant was that the manuscript would benefit from a bit of discussion about glucosaminidases and how they may contribute to septal cleavage in general (so some discussion of septal PG architecture, more detail on the cleavage specificity of a glucosaminidase). This would help the reader put everything into context, and especially help readers who are not familiar with cell wall biochemistry. In addition, it would help people with a Gram-negative background, who typically think of amidases as septal cleavage enzymes.

We have added a short paragraph in the discussion to address the reviewer's suggestion (p. 3, l. 44-46).

2. Western Blots (e.g., Fig. S2BCD) need a loading control. Also, what are bands not marked with an asterisk? Some of these appear to have a positive signal in the zymogram gel (e.g., altA-1-5). Are these activated cleavage product resulting from protease activity?

We have added a loading control in all figures showing immunoblotting.

As stated in the legend, the bands marked with an asterisk correspond to unprocessed AtlA (preprotein). They cannot be "activated cleavage products resulting from protease activity" because their molecular weight is larger than the molecular weight of the full length secreted AtlA protein produced by the WT strain.

I think the authors missed the word "not" in my comment – I was wondering about the bands that were NOT marked with an asterisk. This could be fixed with a simple arrow indicating "non-specific bands" or the likes, but my point stands that some of these might be degradation products.

Apologies for the misunderstanding!

We agree with the reviewer's interpretation. The bands indicated with arrows on the zymogram and Western blot (Fig. S3B and C) correspond to proteolytic products since they display activity on the zymogram. This is clearly indicated in the legend.